# Regularization properties of adversarially-trained linear regression

**Antônio H. Ribeiro**
Uppsala University
`antonio.horta.ribeiro@it.uu.se`

**Dave Zachariah**
Uppsala University
`dave.zachariah@it.uu.se`

**Francis Bach**
PSL Research University, INRIA
`francis.bach@inria.fr`

**Thomas B. Schön**
Uppsala University
`thomas.schon@it.uu.se`

## Abstract

State-of-the-art machine learning models can be vulnerable to very small input perturbations that are adversarially constructed. Adversarial training is an effective approach to defend against it. Formulated as a min-max problem, it searches for the best solution when the training data were corrupted by the worst-case attacks. Linear models are among the simple models where vulnerabilities can be observed and are the focus of our study. In this case, adversarial training leads to a convex optimization problem which can be formulated as the minimization of a finite sum. We provide a comparative analysis between the solution of adversarial training in linear regression and other regularization methods. Our main findings are that: (A) Adversarial training yields the minimum-norm interpolating solution in the overparameterized regime (more parameters than data), as long as the maximum disturbance radius is smaller than a threshold. And, conversely, the minimum-norm interpolator is the solution to adversarial training with a given radius. (B) Adversarial training can be equivalent to parameter shrinking methods (ridge regression and Lasso). This happens in the underparametrized region, for an appropriate choice of adversarial radius and zero-mean symmetrically distributed covariates. (C) For $\ell_\infty$-adversarial training—as in square-root Lasso—the choice of adversarial radius for optimal bounds does not depend on the additive noise variance. We confirm our theoretical findings with numerical examples.

## 1 Introduction

Adversarial attacks generated striking examples of the brittleness of modern machine learning. The framework considers inputs contaminated with disturbances deliberately chosen to maximize the model error and, even for small disturbances, can cause a substantial performance drop in otherwise state-of-the-art models [1]–[6]. Adversarial training [7] is one of the most effective approaches for deep learning models to defend against adversarial attacks [7]–[9]. It considers training models on samples that have been modified by an adversary, with the goal of obtaining a model that will be more robust when faced with new adversarially perturbed samples. The training procedure is formulated as a min-max problem, searching for the best solution to the worst-case attacks.

Despite its success in producing state-of-the-art results in adversarial defense benchmarks [10], there are still major challenges in using adversarial training. The min-max problem is a hard optimization problem to solve and it is still an open question how to solve it efficiently. Moreover, these methods still produce large errors in new adversarially disturbed test points, often much larger than the adversarial error obtained during training.

37th Conference on Neural Information Processing Systems (NeurIPS 2023).

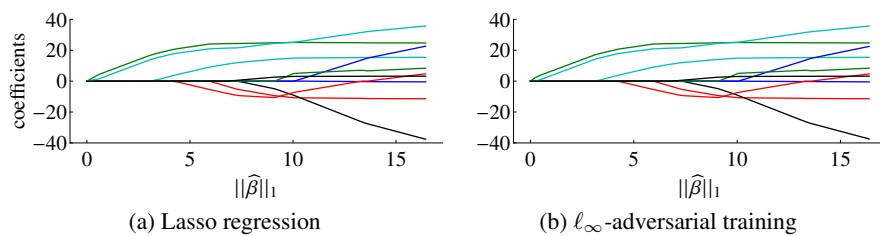

(a) Lasso regression        (b) $\ell_\infty$-adversarial training

Figure 1: **Regularization paths** estimated in the Diabetes dataset [18]. Regularization paths are plots showing the coefficient estimates for varying regularization parameter $\lambda$ or, in adversarial training, perturbation radius $\delta$. This type of plot is commonly used in analysing Lasso and variants, i.e. [18], [19]. On the horizontal axis, we give the $\ell_1$-norm of the estimated parameter. On the vertical axis, the coefficients obtained for each method. The dataset has $p = 10$ baseline variables (age, sex, body mass index, average blood pressure, and six blood serum measurements), which were obtained for $n = 442$ diabetes patients. The model output is a quantitative measure of the disease progression. See Appendix B.5 for the relationship between $\|\widehat{\boldsymbol{\beta}}\|_1$ and $\delta$ and $\lambda$.

To get insight into adversarial training and try to tackle these challenges, a growing body of work [11]–[15] studies fundamental properties of adversarial attacks and adversarial training in linear models. Linear models allow for analytical analysis while still reproducing phenomena observed in state-of-the-art models. Consider a training dataset $\{(\boldsymbol{x}_i, y_i)\}_{i=1}^n$ consisting of $n$ data points of dimension $\mathbb{R}^p \times \mathbb{R}$, adversarial training in linear regression corresponds to minimizing (in $\boldsymbol{\beta} \in \mathbb{R}^p$)

$$R^{\text{adv}}(\boldsymbol{\beta}; \delta, \|\cdot\|) = \frac{1}{n} \sum_{i=1}^n \max_{\|\boldsymbol{\Delta x}_i\| \leq \delta} |y_i - (\boldsymbol{x}_i + \boldsymbol{\Delta x}_i)^\top \boldsymbol{\beta}|^2. \tag{1}$$

Restricting the analysis to linear models allows for a simplified analysis: the problem is convex and the next proposition allows us to express $R^{\text{adv}}$ in terms of the dual norm, $\|\boldsymbol{\beta}\|_* = \sup_{\|\boldsymbol{x}\| \leq 1} |\boldsymbol{\beta}^\top \boldsymbol{x}|$.

> **Proposition 1** (Dual formulation). Let $\|\cdot\|_*$ be the dual norm of $\|\cdot\|$, then
>
> $$R^{\text{adv}}(\boldsymbol{\beta}; \delta, \|\cdot\|) = \frac{1}{n} \sum_{i=1}^n \left(|y_i - \boldsymbol{x}_i^\top \boldsymbol{\beta}| + \delta\|\boldsymbol{\beta}\|_*\right)^2. \tag{2}$$

This simple reformulation removes one of the major challenges: it removes the inner optimization problem and yields a model that is more amenable to analysis. Similar reformulations can be obtained for classification. This simplification made the analysis of adversarial attacks in linear models fruitful to get insights into the properties of adversarial training and examples. From giving a counterexample to the idea that deep neural networks' vulnerabilities to adversarial attacks were caused by their nonlinearity [2], [5], [16], to help explaining how implicitly regularized overparametrized models can be robust, see [17].

In this paper, we further explore this reformulation and *provide a thorough characterization of adversarial training in linear regression problems*. We do this in a comparative fashion, establishing when the solution of adversarially-trained linear models coincides with the solution of traditional regularization methods. For instance, a simple inspection of (2) hints at the similarities with other parameter shrinking methods. This similarity is confirmed in Figure 1,[1] which illustrates how the solutions of $\ell_\infty$-adversarial training can be almost indistinguishable from Lasso. We explain the similarities and differences with other methods. Our contributions are:

A. In the overparametrized region, we prove that the *minimum-norm interpolator is the solution to adversarial training for $\delta$ smaller than a certain threshold* (Section 4).
B. We establish *conditions under which the solution coincides with Lasso and ridge regression*, respectively (Section 5).
C. We show that adversarial training can be framed in the *robust regression* framework, and use it to establish connections with the *square-root Lasso*. We show that adversarial training (like the *square-root Lasso*) can obtain bounds on the prediction error that do not require the noise variance to be known (Section 6).
D. We prove a more *general version of Proposition 1*, valid for general lower-semicontinuous and convex loss functions (Section 8).

---

[1]Code for reproducing the figures is available in: `https://github.com/antonior92/advtrain-linreg`

## 2 Related work

**Generalization of minimum-norm interpolators.** The study of minimum-norm interpolators played a key role in explaining why overparametrized models generalize—an important open question for which traditional theory failed to explain empirical results [20]—and for which significant progress has been made over the past few years [21]. These estimates provide a simple scenario where we can interpolate noisy data and still generalize well. Minimum-norm interpolators have indeed been quite a fruitful setting to study the phenomenon of *benign overfitting*, i.e., when the model interpolates the training data but still generalizes well to new samples. In [22] the authors use these estimates to prove consistency, a development that later had several extensions [23]. From another angle, in a series of insightful papers, Belkin *et al.* [24], [25] explore the phenomena of *double-descent*: where a double-descent curve subsumes the textbook U-shaped bias–variance trade-off curve, with a second decrease in the error occurring beyond the point where the model has reached the capacity of interpolating the training data. Interestingly, minimum-norm interpolators are also basic scenarios for observing this phenomena [24]–[26]. *Our research connects the research on robustness to adversarial attacks to the study of generalization of minimum-norm interpolators.*

**Adversarial training in linear models.** The generalization of adversarial attacks in linear models is well-studied. Tsipras *et al.* [16] and Ilyas *et al.* [5] use linear models to explain the conflict between robustness and high-performance models observed in neural networks; Ribeiro *et al.* [17] use these models to show how overparameterization affects robustness to perturbations; Taheri *et al.* [11] derive asymptotics for adversarial training in binary classification. Dan *et al.* [27] and Dobriban *et al.* [28] study adversarial robustness in Gaussian classification problems. Javanmard *et al.* [12] provide asymptotics for adversarial training in linear regression, Javanmard *et al.* [29] in classification settings and Hassani *et al.* [13] for random feature regressions. Min *et al.* [14] investigate how the dataset size affects adversarial performance. Yin *et al.* [15] provide an analysis of $\ell_\infty$-attack on linear classifiers based on Rademacher complexity. These works, however, focus on the generalization properties. *We provide a fresh perspective on the problem by clarifying the connection of adversarial training to other regularization methods.*

**Robust regression and square-root Lasso.** The properties of Lasso [30] for recovering sparse parameters in noisy settings have been extensively studied, and bounds on the parameter estimation error are well-known [31]. However, these results rely on choices of regularization parameters that depend on the variance of additive noise. Square-root Lasso [32] was proposed to circumvent this difficulty. We show that for $\ell_\infty$-adversarial, *similarly to the square-root Lasso*, bounds can be obtained with the adversarial radius set without knowledge about the variance of the additive noise. We also show that both methods fit into the robust regression framework [33]. The work on robust classification [34] is also connected to our work, there they show that certain robust support vector machines are equivalent to SVM. Our results for classification in Section 8 can be viewed as a generalization of their Theorem 3.

**Dual formulation.** Variations of Proposition 1 are presented in [12], [17], [35]. Equivalent results in the context of classification are presented in [2], [15]. We use the reformulation extensively in our developments and also present a generalized statement for it.

## 3 Background

Different estimators will be relevant to our developments and will be compared in this paper.

**Adversarially-trained linear regression.** Our main object of study is the minimization of

$$R^{\text{adv}}(\boldsymbol{\beta}; \delta, \|\cdot\|) = \frac{1}{n} \sum_{i=1}^{n} \max_{\|\boldsymbol{\Delta x}_i\| \leq \delta} |y_i - (\boldsymbol{x}_i + \boldsymbol{\Delta x}_i)^\top \boldsymbol{\beta}|^2 \overset{(a)}{=} \frac{1}{n} \sum_{i=1}^{n} \left( |y_i - \boldsymbol{x}_i^\top \boldsymbol{\beta}| + \delta \|\boldsymbol{\beta}\|_* \right)^2,$$

where equality (a) follows from Proposition 1. We use $\|\boldsymbol{\beta}\|_* = \sup_{\|\boldsymbol{x}\| \leq 1} |\boldsymbol{\beta}^\top \boldsymbol{x}|$ to denote the dual norm of $\|\cdot\|$. We highlight that the $\ell_2$-norm, $\|\boldsymbol{\beta}\|_2 = \sum_i |\boldsymbol{\beta}_i|^2$ is dual to itself. The $\ell_1$-norm, $\|\boldsymbol{\beta}\|_1 = \sum_i |\boldsymbol{\beta}_i|$, is the dual norm of the $\ell_\infty$-norm, $\|\boldsymbol{\beta}\|_\infty = \max_i |\boldsymbol{\beta}_i|$. When the adversarial disturbances are constrained to the $\ell_p$ ball: $\{\boldsymbol{\Delta x} : \|\boldsymbol{\Delta x}\|_p \leq \delta\}$ we will call it $\ell_p$-adversarial training. Our focus will be on $\ell_\infty$ and $\ell_2$-adversarial training.

**Parameter-shrinking methods.** Parameter-shrinking methods explicitly penalize large values of $\boldsymbol{\beta}$. Regression methods that involve shrinkage include Lasso [30] and ridge regression, which minimize,

respectively

$$R^{\text{lasso}}(\boldsymbol{\beta}; \lambda) = \frac{1}{n} \sum_{i=1}^{n} |y_i - \boldsymbol{x}_i^\top \boldsymbol{\beta}|^2 + \lambda \|\boldsymbol{\beta}\|_1 \quad \text{and} \quad R^{\text{ridge}}(\boldsymbol{\beta}; \lambda) = \frac{1}{n} \sum_{i=1}^{n} |y_i - \boldsymbol{x}_i^\top \beta|^2 + \lambda \|\boldsymbol{\beta}\|_2^2.$$

**Square-root Lasso:** Setting (with guarantees) the Lasso regularization parameters requires knowledge about the variance of the noise. Square-root Lasso [32] circumvent this difficulty by minimizing:

$$R^{\sqrt{\text{lasso}}}(\boldsymbol{\beta}, \lambda) = \sqrt{\frac{1}{n} \sum_{i=1}^{n} |y_i - \boldsymbol{x}_i^\top \boldsymbol{\beta}|^2} + \lambda \|\boldsymbol{\beta}\|_1. \tag{3}$$

**Minimum-norm interpolator:** Let $\boldsymbol{X} \in \mathbb{R}^{n \times p}$ denote the matrix of stacked vectors $\boldsymbol{x}_i$ and $\boldsymbol{y} \in \mathbb{R}^n$ is the vector of stacked outputs. When matrix $\boldsymbol{X}$ has full row rank, the linear system $\boldsymbol{X}\boldsymbol{\beta} = \boldsymbol{y}$ has multiple solutions. In this case the minimum $\| \cdot \|_*$-norm interpolator is the solution of

$$\min_{\boldsymbol{\beta}} \|\boldsymbol{\beta}\|_* \quad \text{subject to} \quad \boldsymbol{X}\boldsymbol{\beta} = \boldsymbol{y}. \tag{4}$$

# 4   Adversarial training in the overparametrized regime

Our first contribution is to provide conditions for when the minimum-norm interpolator is equivalent to adversarial training in the overparametrized case (we will assume that rank$(\boldsymbol{X}) = n$ and $n < p$). On the one hand, our result gives further insight into adversarial training. On the other hand, it allows us to see minimum-norm interpolators as a solution of the adversarial training problem.

> **Theorem 1.** Assume the matrix $\boldsymbol{X} \in \mathbb{R}^{n \times p}$ to have full row rank, let $\widehat{\boldsymbol{\alpha}}$ denote the solution of the dual problem $\max_{\|\boldsymbol{\alpha}^\top \boldsymbol{X}\| \leq 1} \boldsymbol{\alpha}^\top \boldsymbol{y}$, and $\bar{\delta}$ denote the threshold
>
> $$\bar{\delta} = (n\|\widehat{\boldsymbol{\alpha}}\|_\infty)^{-1}. \tag{5}$$
>
> The minimum $\| \cdot \|_*$-norm interpolator minimizes the adversarial risk $R^{\text{adv}}(\boldsymbol{\beta}, \delta, \| \cdot \|)$ if and only if $\delta \in (0, \bar{\delta}]$.

*Remark* 1 (Bounds on $\bar{\delta}$). The theorem allows us to directly compute $\bar{\delta}$, but it does require solving the dual problem to the minimum-norm solution. In Appendix A.3, we provide bounds on $\bar{\delta}$ that avoid solving the optimization problem. For instance, for $\ell_\infty$-adversarial attacks, we have the following bounds depending on the singular values of $\boldsymbol{X}$: $\frac{1}{\sqrt{p}} \sigma_n(\boldsymbol{X}) \leq n\bar{\delta} \leq \sqrt{p}\sigma_1(\boldsymbol{X})$. Where $\sigma_1$ and $\sigma_n$ denote the largest and smallest *positive* singular values of $\boldsymbol{X}$, respectively.

*Proof of Theorem 1.* Let $\widehat{\boldsymbol{\beta}}$ and $\widehat{\boldsymbol{\alpha}}$ be the minimum $\| \cdot \|_*$-norm solution and the solution of the associated dual problem. Throughout the proof we denote $R^{\text{adv}}(\boldsymbol{\beta}) = R^{\text{adv}}(\boldsymbol{\beta}; \delta, \| \cdot \|)$, dropping the last two arguments. The subgradient of $R^{\text{adv}}(\boldsymbol{\beta})$ evaluated at $\widehat{\boldsymbol{\beta}}$ is given by[2]

$$\partial R^{\text{adv}}(\widehat{\boldsymbol{\beta}}) = \frac{2\delta}{n} \sum_{i=1}^{n} \|\widehat{\boldsymbol{\beta}}\|_* (\rho_i \boldsymbol{x}_i + \delta \partial \|\widehat{\boldsymbol{\beta}}\|_*), \tag{6}$$

for any $\boldsymbol{\rho} = (\rho_1, \ldots, \rho_n) \in \mathbb{R}^n$ that satisfies $\|\boldsymbol{\rho}\|_\infty \leq 1$. We have that any element of $\partial \|\widehat{\boldsymbol{\beta}}\|_*$ can be written as $\boldsymbol{X}^\top \widehat{\boldsymbol{\alpha}}$ (we prove that in Lemma 1 in the Appendix). Hence, we can rewrite

$$\partial R^{\text{adv}}(\widehat{\boldsymbol{\beta}}) = \frac{2\delta}{n} \sum_{i=1}^{n} \|\widehat{\boldsymbol{\beta}}\|_* (\rho_i \boldsymbol{x}_i + \delta \boldsymbol{X}^T \widehat{\boldsymbol{\alpha}}) = 2\delta \|\widehat{\boldsymbol{\beta}}\|_* \left( \frac{\boldsymbol{X}^\top \boldsymbol{\rho}}{n} + \delta \boldsymbol{X}^\top \widehat{\boldsymbol{\alpha}} \right).$$

If $\delta \leq \bar{\delta} = 1/(n\|\widehat{\boldsymbol{\alpha}}\|_\infty)$, we can take $\boldsymbol{\rho} = -n\delta\widehat{\boldsymbol{\alpha}}$ and the subderivative contains zero. On the other hand, if $\delta > 1/(n\|\widehat{\boldsymbol{\alpha}}\|_\infty)$ then $\left( \frac{\boldsymbol{X}^\top \boldsymbol{\rho}}{n} + \delta \boldsymbol{X}^\top \widehat{\boldsymbol{\alpha}} \right)$ is not zero for $\|\boldsymbol{\rho}\|_\infty \leq 1$. $\qquad \square$

The minimum $\ell_1$-norm interpolator is well studied in the context of 'basis pursuit' and allows the recovery of low-dimensional representations in sparse signals [39]. The interest in the minimum $\ell_2$-norm is more recent. It played an important role in studying the interplay between interpolation and generalization, being used in many recent papers where the double-descent [24], [26] and the benign overfitting phenomena [22] were observed.

---

[2]The subgradient of a function $\omega : \mathbb{R}^p \to \mathbb{R}$ is the set:

$$\partial \omega(\boldsymbol{\beta}_0) = \{\boldsymbol{v} \in \mathbb{R}^p : \omega(\boldsymbol{\beta}) - \omega(\boldsymbol{\beta}_0) \geq \boldsymbol{v}(\boldsymbol{\beta} - \boldsymbol{\beta}_0), \forall \boldsymbol{\beta} \in \mathbb{R}^p\}.$$

See [36]–[38] for properties. We use $\partial \|\widehat{\boldsymbol{\beta}}\|_*$ to denote the subgradient of $\omega(\boldsymbol{\beta}) = \|\boldsymbol{\beta}\|_*$ evaluated at $\widehat{\boldsymbol{\beta}}$.

Theorem 1 gives a new perspective on the robustness of the minimum-norm interpolator, allowing us to see it as a solution to adversarial training with adversarial radius $\bar{\delta}$. Figure 2 shows that the radius $\bar{\delta}$ of the adversarial training problem corresponding to the minimum-norm interpolator increases with the ratio $p/n$. It is natural to expect that training with a larger radius would yield more adversarially robust models on new test points. The next result formalizes this intuition by establishing an upper bound on the robustness gap: the difference between the square-root of the expected adversarial squared error

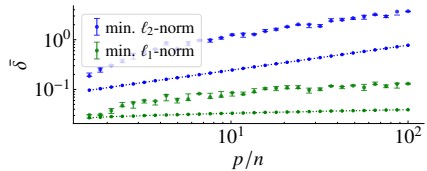

Figure 2: **Threshold $\bar{\delta}$ vs. number of features.** We fix $n = 60$ and show the value of $\bar{\delta}$ (defined in (5)) as a function of the number of features $p$. The dotted lines give the reference $\delta_{\text{test}} = 0.01\mathbb{E}\left[\|\boldsymbol{x}\|\right]$ for comparison.

$\mathcal{R}_*^{\text{adv}}(\boldsymbol{\beta}; \delta_{\text{test}}, \|\cdot\|) = \mathbb{E}_{y_0, \boldsymbol{x}_0}\left[\max_{\|\boldsymbol{\Delta}\boldsymbol{x}_i\| \leq \delta_{\text{test}}}(y_0 - (\boldsymbol{x}_0 + \boldsymbol{\Delta}\boldsymbol{x}_0)^\top\boldsymbol{\beta})^2\right]$ and the expected squared error $\mathcal{R}_*(\boldsymbol{\beta}) = \mathbb{E}_{y_0, \boldsymbol{x}_0}\left[(y_0 - \boldsymbol{x}_0^\top\boldsymbol{\beta})^2\right]$. The upper bound depends on the in-training adversarial error and on the ratio $\frac{\delta_{\text{test}}}{\bar{\delta}}$ which are quantities that you can directly compute.

> **Proposition 2.** Assume $\boldsymbol{X}$ to have full row rank and let $\widehat{\boldsymbol{\beta}}$ be the minimum $\|\cdot\|_*$-norm interpolator, than
>
> $$\sqrt{\mathcal{R}_*^{\text{adv}}(\widehat{\boldsymbol{\beta}}; \delta_{\text{test}}, \|\cdot\|)} - \sqrt{\mathcal{R}_*(\widehat{\boldsymbol{\beta}})} \leq \frac{\delta_{\text{test}}}{\bar{\delta}}\sqrt{R^{\text{adv}}(\widehat{\boldsymbol{\beta}}; \bar{\delta}, \|\cdot\|)}. \tag{7}$$

*Remark* 2. The example in Figure 2 is one case where adding more features makes the minimum-norm interpolator more robust. Examples showing the opposite and illustrating that adding more features to linear models can make them less adversarially robust are abound in the literature [16], [17]. Indeed, examples that consider the minimum $\ell_2$-interpolator subject to $\ell_\infty$-adversarial attacks during test-time result in this scenario, as discussed in [17]. Proposition 7 in the Appendix is the equivalent of Proposition 2 for this case (mismatched norms in train and test). There, the upper bound grows with $\sqrt{p}$, which explains the vulnerability in this scenario.

*Remark* 3. A pitfall of analyzing the minimum-norm solution properties in linear models is that it requires the analysis of nested problems: different choices of $p$ require different covariates $\boldsymbol{x}$. The random projection model proposed by Bach [40] avoids this pitfall by considering the input $\boldsymbol{x}$ is fixed, but only a projected version of it $\boldsymbol{S}\boldsymbol{x}$ is observed, with the number of parameters being estimated changing with the number of projections observed. Adversarially training this model consists in finding a parameter $\widehat{\boldsymbol{\beta}}$ that minimizes: $\frac{1}{n}\sum_{i=1}^n \max_{\|\boldsymbol{\Delta}\boldsymbol{x}_i\| \leq \delta}|y_i - (\boldsymbol{x}_i + \boldsymbol{\Delta}\boldsymbol{x}_i)^\top\boldsymbol{S}^\top\boldsymbol{\beta}|^2$. We generalize our results so that we can also study this case (See Appendix A.4).

## 5 Adversarial training and parameter-shrinking methods

Theorem 1 stated an equivalence with the minimum-norm solution when $\delta$ is small. The next result applies to the other extreme of the spectrum, characterizing the case when $\delta$ is large.

> **Proposition 3** (Zero solution of adversarial training). The zero solution $\widehat{\boldsymbol{\beta}} = \boldsymbol{0}$ minimizes the adversarial training if and only if $\delta \geq \frac{\|\boldsymbol{X}^\top\boldsymbol{y}\|}{\|\boldsymbol{y}\|_1}$.

In the remaining of this section, we will characterize the solution between these two extremes. We compare adversarial training to Lasso and ridge regression.

### 5.1 Relation to Lasso and ridge regression

Proposition 1 makes it clear by inspection that adversarial training is similar to other well-known parameter-shrinking regularization methods. Indeed, for $\ell_\infty$-adversarial attacks, the cost function $R^{\text{adv}}(\beta; \delta, \|\cdot\|_\infty)$ in its dual form is remarkably similar to Lasso [30] and $R^{\text{adv}}(\beta; \delta, \|\cdot\|_2)$, to ridge regression (the cost functions were presented in Section 3). In Figures 1 and 3, we show the regularization paths of these methods (the dataset is described in [18]). We observe that $\ell_\infty$-*adversarial training produces sparse solutions* and that Lasso and $\ell_\infty$-adversarial training have extremely similar regularization paths. There are also striking similarities between ridge regression and $\ell_2$-adversarial training, with the notable difference that for large $\delta$ the solution of $\ell_2$-adversarial

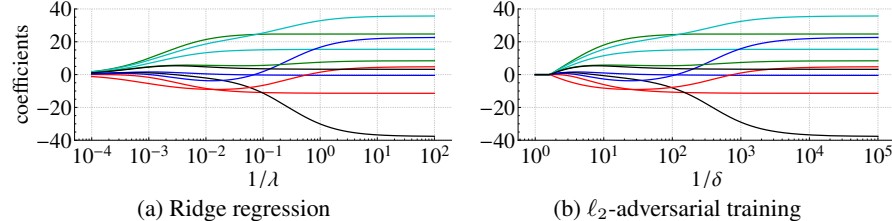

(a) Ridge regression        (b) $\ell_2$-adversarial training

Figure 3: **Regularization paths** in the Diabetes dataset [18] (see dataset description in Figure 1). On the horizontal axis, we give the inverse of the regularization parameter (in log scale). On the vertical axis, the coefficients.

training is zero (which is explained by Proposition 3). The next proposition justifies why we can observe such similarities in part of the regularization path. It applies to data that has been normalized and for positive responses $y$ (as it is the case in our example).

> **Proposition 4.** Assume the output is positive and the data is normalized: $\boldsymbol{y} \geq 0$ and $\boldsymbol{X}^\top \mathbf{1} = \mathbf{0}$. The solution of adversarial training $\widehat{\boldsymbol{\beta}}$, for $\|\widehat{\boldsymbol{\beta}}\|_* \leq \min_i \frac{|y_i|}{\|x_i\|}$ is also the solution of
> $$\min_{\boldsymbol{\beta}} \frac{1}{n} \sum_{i=1}^n |y_i - \boldsymbol{x}_i^\top \boldsymbol{\beta}|^2 + \left(\delta\|\boldsymbol{\beta}\|_* + \tfrac{1}{n}\|\boldsymbol{y}\|_1\right)^2.$$

The proposition is proved in Appendix B. It establishes the equivalence between $\ell_\infty$-adversarial training and Lasso (even though there is not a closed-formula expression for the map between $\delta$ and $\lambda'$). Indeed, *under the assumptions of the propositions, $\widehat{\boldsymbol{\beta}}$ is the solution of $\ell_\infty$-adversarial problem only if it is the solution of*

$$\min_{\beta} \frac{1}{n} \sum_{i=1}^n (y_i - \boldsymbol{x}_i^\top \boldsymbol{\beta}) + \lambda'\|\boldsymbol{\beta}\|_1, \tag{8}$$

*for some $\lambda'$.* We can prove this statement using the proposition: under the appropriate assumptions, the $\ell_\infty$-adversarial problem solution is also a solution to the problem

$$\min_{\boldsymbol{\beta}} \frac{1}{n} \sum_{i=1}^n (y_i - \boldsymbol{x}_i^\top \boldsymbol{\beta})^2 + \left(\delta\|\boldsymbol{\beta}\|_1 + \tfrac{1}{n}\|\boldsymbol{y}\|_1\right)^2,$$

which in turn, is the Lagrangian formulation of the following constrained optimization problem

$$\min_{\boldsymbol{\beta}} \frac{1}{n} \sum_{i=1}^n (y_i - \boldsymbol{x}_i^\top \boldsymbol{\beta})^2 \text{ subject to } \left(\delta\|\boldsymbol{\beta}\|_1 + \tfrac{1}{n}\|\boldsymbol{y}\|_1\right)^2 \leq \Delta,$$

for some $\Delta \geq \frac{1}{n}\|\boldsymbol{y}\|_1$. The constraint can be rewritten as $\delta\|\boldsymbol{\beta}\|_1 \leq \sqrt{\Delta} - \frac{1}{n}\|\boldsymbol{y}\|_1$. And, the result follows because (8) is the Lagrangian formulation of this modified problem. A similar reasoning could be used to connect the result for $\ell_2$-adversarial training with ridge regression.

More generally, even when the condition on $\widehat{\boldsymbol{\beta}}$ is not satisfied and if $\boldsymbol{y}$ is negative, we show in Appendix B that for zero-mean and symmetrically distributed covariates, i.e., $\mathbb{E}[\boldsymbol{x}] = 0$ and $(\boldsymbol{x} \sim -\boldsymbol{x})$, adversarial training has approximately the same solution as

$$\min_{\boldsymbol{\beta}} \frac{1}{n} \sum_{i=1}^n |y_i - \boldsymbol{x}_i^\top \boldsymbol{\beta}|^2 + \left(\delta\|\boldsymbol{\beta}\|_* + \tfrac{1}{n}\boldsymbol{s}^\top \boldsymbol{y}\right)^2,$$

for $\boldsymbol{s} = \text{sign}(\boldsymbol{y} - \boldsymbol{X}^\top \widehat{\boldsymbol{\beta}})$. This explains the similarities in the regularization paths.

### 5.2 Transition into the interpolation regime

The discussion above hints at the similarities between adversarial training and parameter-shrinking methods. Interestingly, Lasso and ridge regression are also connected to minimum-norm interpolators. The ridge regression solution converges to the minimum-norm solution as the parameter vanishes

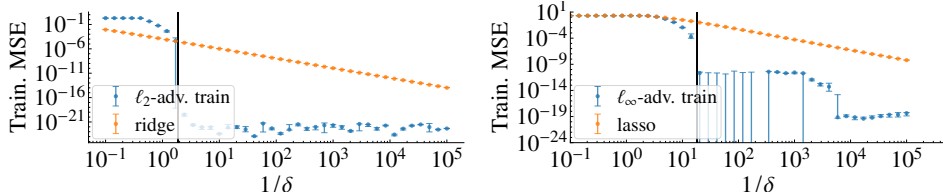

Figure 4: **Training mean squared error *vs* inverse adversarial radius**. *Left:* for ridge and $\ell_2$-adversarial training. *Right:* for Lasso and $\ell_\infty$-adversarial training. The error bars give the median and the 0.25 and 0.75 quantiles from 5 realizations. The vertical black lines show $\bar{\delta}$ in (5). Figure S.9 (appendix) shows the test MSE. For ridge regression or Lasso (see Section 3), the $x$-axis should be read as $1/\lambda$, rather than $1/\delta$. We generate the data synthetically using an isotropic Gaussian feature model (see Section 7) with $n = 60$ training data points and $p = 200$ features.

i.e., $\widehat{\boldsymbol{\beta}}^{\text{ridge}}(\lambda) \to \widehat{\boldsymbol{\beta}}^{\text{min}-\ell_2}$ as $\lambda \to 0^+$. Similarly, there is a relation between the minimum $\ell_1$-norm solution and Lasso. The relation requires additional constraints because, for the overparameterized case, Lasso does not necessarily have a unique solution. Nonetheless, it is proved in [41, Lemma 7] that the Lasso solution by the LARS algorithm satisfies $\widehat{\boldsymbol{\beta}}^{\text{lasso}}(\lambda) \to \widehat{\boldsymbol{\beta}}^{\text{min}-\ell_1}$ as $\lambda \to 0^+$.

For a sufficiently small $\delta$, the solution of $\ell_2$-adversarial training equals the minimum $\ell_2$-norm solution; and, the solution of $\ell_\infty$-adversarial training equals the minimum $\ell_1$-norm solution. There is a notable difference though: while this happens *only in the limit* for ridge regression and Lasso, for adversarial training this happens *for all values $\delta$ smaller than* the threshold $\bar{\delta}$. We illustrate this phenomenon next. Unlike ridge regression and Lasso which converge towards the interpolation solution, adversarial training goes through abrupt transitions and suddenly starts to interpolate the data. Figure 4 illustrates this phenomenon in synthetically generated data (with isotropic Gaussian feature, see Section 7).

**Intuitive explanation for abrupt transitions.** To get insight into why the abrupt transition occurs we present the case where $n = 1$, i.e., the dataset has a single data point $(\boldsymbol{x}, y)$, where $y = 1$ and $\|\boldsymbol{x}\|_2 = 1$. We can write $R^{\text{adv}}(\boldsymbol{\beta}; \delta, \|\cdot\|_2) = (\mathcal{L}(\boldsymbol{\beta}))^2$, where

$$\mathcal{L}(\boldsymbol{\beta}) = |y - \boldsymbol{x}^\mathsf{T}\boldsymbol{\beta}| + \delta\|\boldsymbol{\beta}\|_2.$$

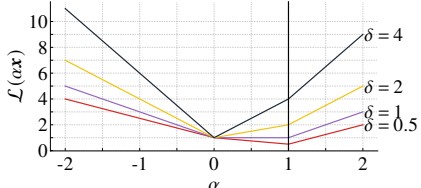

Figure 5: **Function** $\mathcal{L}(\boldsymbol{\beta})$ for $\boldsymbol{\beta} = \alpha\boldsymbol{x}$.

The function $\mathcal{L}$ is necessarily minimized along the subspace spanned by the vector $\boldsymbol{x}$. Now, along this line, the function $\mathcal{L}$ is piecewise linear with three segments, see Figure 5. One of the three segments changes the slope sign as $\delta$ decreases and the minimum of the function changes abruptly, thus explaining why abrupt transitions occur.

## 6 Relation to robust regression and square-root Lasso

In this section we establish that both adversarial training and square-root Lasso can be integrated into the robust regression framework. We further investigate the similarities between the methods by analyzing the prediction error upper bounds.

**Robust regression framework.** Robust linear regression considers the minimization of the following cost function

$$R^{\text{robust}}(\boldsymbol{\beta}; \mathcal{S}) = \max_{\boldsymbol{\Delta} \in \mathcal{S}} \|\boldsymbol{y} - (\boldsymbol{X} + \boldsymbol{\Delta})\boldsymbol{\beta}\|_2, \tag{9}$$

where the disturbance matrix $\boldsymbol{\Delta}$ is constrained to belong to the 'disturbance set' $\mathcal{S}$. We connect robust regression, square-root Lasso and adversarial training. The next proposition gives the equivalence between robust regression for a row-bounded disturbance sets $\mathcal{R}_{p,\delta}$ and $\ell_p$-adversarial training, see the appendix for the proof.

**Proposition 5.** For a disturbance set with the rows bounded by $\delta$:

$$\mathcal{R}_{p,\delta} = \left\{ \begin{bmatrix} \text{---} & \boldsymbol{\Delta x}_1 & \text{---} \\ & \vdots & \\ \text{---} & \boldsymbol{\Delta x}_n & \text{---} \end{bmatrix} : \|\boldsymbol{\Delta x}_i\| \le \delta, \forall i \right\},$$

we have that $\arg\min_{\boldsymbol{\beta}} R^{\text{robust}}(\boldsymbol{\beta}, \mathcal{R}_{p,\delta}) = \arg\min_{\boldsymbol{\beta}} R^{\text{adv}}(\boldsymbol{\beta}; \delta, \|\cdot\|)$.

On the other hand, there is an equivalence between square-root Lasso and robust regression for column-bounded disturbance sets $\mathcal{C}_{2,\delta}$. This was established by Xu *et al.* [33, Theorem 1] and we repeat it below.

**Proposition 6.** [33] For a disturbance set with columns bounded by $\delta$:

$$\mathcal{C}_{2,\delta} = \left\{ \begin{bmatrix} | & & | \\ \boldsymbol{\zeta}_1 & \cdots & \boldsymbol{\zeta}_m \\ | & & | \end{bmatrix} : \|\boldsymbol{\zeta}_i\|_2 \le \delta, \forall i \right\},$$

we have that $\arg\min_{\boldsymbol{\beta}} R^{\text{robust}}(\boldsymbol{\beta}; \mathcal{C}_{2,\delta}) = \arg\min_{\boldsymbol{\beta}} R^{\sqrt{\text{lasso}}}(\boldsymbol{\beta}, \delta)$.

The above discussion hints at the similarities between adversarial training and square-root Lasso, as instances of robust regression under different constraints. Square-root Lasso minimizes $R^{\sqrt{\text{lasso}}}(\boldsymbol{\beta}, \lambda) = n^{-1/2}\|\boldsymbol{y} - \boldsymbol{X}\boldsymbol{\beta}\|_2 + \lambda\|\boldsymbol{\beta}\|_1$ (See Section 3). The main motivation for the square-root Lasso is that it is a 'pivotal' method for sparse recovery: that is, it attains near-oracle performance without knowledge of the variance levels to set the regularization parameter [32]. As we will show in the next section, a similar property applies to $\ell_\infty$-adversarial training.

**Fixed-design analysis and similarities with square-root Lasso.** In this section, we assume that the data was generated as: $y_i = \boldsymbol{x}_i^\top \boldsymbol{\beta}^* + \varepsilon_i$ where $\boldsymbol{\beta}^*$ is the parameter vector used to generate the data. Under these assumptions, we can derive an upper bound for the (in-sample) prediction error:

**Theorem 2.** Let $\delta > \delta^* = 3\frac{\|\boldsymbol{X}^\top \boldsymbol{\varepsilon}\|_\infty}{\|\boldsymbol{\varepsilon}\|_1}$, the prediction error of $\ell_\infty$-adversarial training satisfies the bound:

$$\frac{1}{n}\|\boldsymbol{X}(\widehat{\boldsymbol{\beta}} - \boldsymbol{\beta}^*)\|_2^2 \le 8\delta\|\boldsymbol{\beta}^*\|_1 \left(\frac{1}{n}\|\boldsymbol{\varepsilon}\|_1 + 10\delta\|\boldsymbol{\beta}^*\|_1\right). \tag{10}$$

For comparison, we also provide the result for Lasso: (Adapted from [31, Thm. 7.13, p. 210])

**Theorem 3.** [31] Let $\lambda > \lambda^* = 3\|\frac{\boldsymbol{X}^\top \boldsymbol{\varepsilon}}{n}\|_\infty$, the prediction error of Lasso satisfies the bound:

$$\frac{1}{n}\|\boldsymbol{X}(\widehat{\boldsymbol{\beta}} - \boldsymbol{\beta}^*)\|_2^2 \le 8\lambda\|\boldsymbol{\beta}^*\|_1. \tag{11}$$

Without additional constraints (see Remark 5) it can be shown that for Lasso it is not possible to improve the above bound. To satisfy this bound, however, Lasso requires knowledge of the magnitude of the noise $\boldsymbol{\varepsilon}$. In Theorem 3, $\lambda^* = \frac{3}{n}\|\boldsymbol{X}^\top \boldsymbol{\varepsilon}\|$. Hence, if we rescale $\boldsymbol{\varepsilon}$ (i.e., $\boldsymbol{\varepsilon} \to \eta\boldsymbol{\varepsilon}$) then a correspondent change in magnitude follows in $\lambda^*$ (i.e., $\lambda^* \to \eta\lambda^*$). Square-root Lasso [32] avoids this problem and achieves a similar rate even without knowing the variance: i.e., it is a 'pivotal' method. Our method has similar properties and allows us to set $\delta$ without estimating the variance. This can be seen in Theorem 2, where re-scaling $\boldsymbol{\varepsilon}$ does not alter the value of $\delta^* = 3\|\boldsymbol{X}^\top \boldsymbol{\varepsilon}\|/\|\boldsymbol{\varepsilon}\|_1$ because it affects the numerator and denominator simultaneously.

For instance, if we assume $\boldsymbol{\varepsilon}$ has i.i.d. $\mathcal{N}(0, \sigma^2)$ entries and that the matrix $\boldsymbol{X}$ is fixed with $\max_{j=1,\dots,m} \|\boldsymbol{x}_j\|_\infty \le M$. For $\lambda \propto M\sigma\sqrt{(\log p)/n}$, we (with high-probability) satisfy the condition in Theorem 3, obtaining: $\frac{1}{n}\|\boldsymbol{X}(\widehat{\boldsymbol{\beta}} - \boldsymbol{\beta}^*)\|_2^2 \lesssim M\sigma\sqrt{(\log p)/n}$. For adversarial training, we can set: $\delta \propto M\sqrt{(\log p)/n}$, and (with high-probability) satisfy the theorem condition, obtaining the same bound. Notice that the choice of $\delta$ is not dependent on $\sigma$. We provide a full analysis in Appendix C.3.

*Remark 4* (On the relation between $\bar{\delta}$ and $\delta^*$). For sufficiently large $n$, we have $\delta^* > \bar{\delta}$ in the scenario above—the noise $\boldsymbol{\varepsilon}$ has i.i.d. normal entries $\mathcal{N}(0, \sigma^2)$ and the matrix $\boldsymbol{X}$ is fixed. We can prove it by contradiction: if $\delta^* \le \bar{\delta}$ then we could apply the bound from Theorem 2 to the minimum $\ell_1$-norm

interpolator (due to Theorem 1). Hence, $\sigma^2 \approx \frac{1}{n}\|\varepsilon\|_2^2 = \frac{1}{n}\|\boldsymbol{X}(\widehat{\boldsymbol{\beta}} - \boldsymbol{\beta}^*)\|_2^2 \lesssim M\sigma\sqrt{(\log p)/n}$ and we can always choose a sufficiently large value of $n$ for which the inequality is false.

*Remark* 5. In the original square-root Lasso paper [32], a bound is obtained for the $\ell_2$ parameter distance: $\|\widehat{\boldsymbol{\beta}} - \boldsymbol{\beta}^*\|_2^2$ for the case $\boldsymbol{X}$ satisfy the restricted eigenvalue condition and $\boldsymbol{\beta}^*$ is sparse. Under these more strict assumptions (restricted eigenvalue condition) we can also obtain a faster convergence rate for the prediction error. Here we focus on the fixed-design prediction error without additional assumptions on $\boldsymbol{X}$. For other predictors, these other proofs follow similar steps and yield similar requirements on $\lambda^*$, see [31, Chapter 7]. However, we leave these and other analysis (such as variable selection consistency) of adversarial training for future work.

# 7 Numerical Experiments

We study five different examples. The main goal is to experimentally confirm our theoretical findings. For each scenario, we compute and plot: train and test MSE for different choices of $p$, $n$ and $\delta$. For comparison purposes, we also compute and plot train and test MSE for Lasso and ridge regression and minimum-norm interpolators. Finally, in line with our discussion in Section 4, we compute and plot $\bar{\delta}$ as a function of $p/n$. In all the numerical examples the adversarial training solution is implemented by minimizing (2) using CVXPY [42]. The scenarios under consideration are described below (see Appendix D for additional details)

1. **Isotropic Gaussian feature model**. The output is a linear combination of the features plus additive noise: $y_i = \boldsymbol{x}_i^\top \boldsymbol{\beta} + \epsilon_i$, for Gaussian noise and covariates: $\epsilon_i \sim \mathcal{N}(0, \sigma^2)$ and $\boldsymbol{x}_i \sim \mathcal{N}(0, \boldsymbol{I}_p)$.
2. **Latent-space feature model** [26, Section 5.4]. The features $\boldsymbol{x}$ are noisy observations of a lower-dimensional subspace of dimension $d$. A vector in this *latent space* is represented by $\boldsymbol{z} \in \mathbb{R}^d$. $\boldsymbol{x} = \boldsymbol{W}\boldsymbol{z} + \boldsymbol{u}$. The output is a linear combination of the latent space plus noise.
3. **Random Fourier features model** [43]. We apply a random Fourier feature map to inputs of the Diabetes dataset [18]. Random fourier features are obtained by the transformation $\boldsymbol{x}_i = \sqrt{2/m}\cos(\boldsymbol{W}\boldsymbol{z}_i + \boldsymbol{b})$ where the entries of $\boldsymbol{W}$ and $\boldsymbol{b}$ are independently sampled. It can be seen as a one-layer untrained neural network and approximates the Gaussian kernel feature map [43].
4. **Random projection model** [40]. The data is generated as in the isotropic Gaussian feature model, but we only observe $\boldsymbol{S}\boldsymbol{x}_i$, i.e., a projection of the inputs.
5. **Phenotype prediction from genotype.** We illustrate our method on the Diverse MAGIC wheat dataset [44] from the National Institute for Applied Botany. We use a subset of the genotype to predict one of the continuous phenotypes.

**Results.** We experimentally corroborate Theorem 1 in all scenarios: In Figures 4 and S.8 (Appendix) we show the train MSE as we change the adversarial radius $\delta$, confirming the abrupt transitions into the interpolation regime and also that $\bar{\delta}$ is the transition point. In Figures S.9 and S.10 we show the corresponding test MSE without and in the presence of an adversary, respectively. Interestingly, the adversarial radius that yields the best results is not always equal to the radius $\delta_{\text{test}}$ the model will be evaluated on.

Figures 2 and S.5 (Appendix) display $\bar{\delta}$ as a function of the ratio $p/n$. We observe that $\bar{\delta}/\mathbb{E}\left[\|\boldsymbol{x}\|\right]$ is growing in all examples and we would still expect improved robustness in light of Proposition 2. The Random Fourier features model is the only case where $\bar{\delta}$ seems to decrease (in absolute value). However, $\mathbb{E}\left[\|\boldsymbol{x}\|_1\right]$ is also decreasing (and at a faster rate). Figures S.6 and S.7 give test MSE without and in the presence of an adversary for the minimum-norm interpolator as a function of the ratio $p/n$, confirming improved adversarial robustness as $p$ grows.

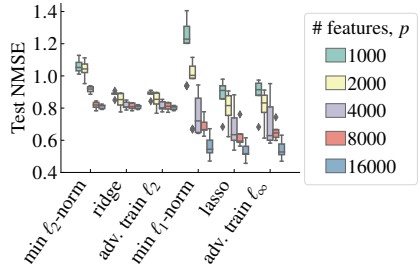

Figure 6: **Test normalized MSE (NMSE)** in the MAGIC dataset.

Figure 6 provides a comparison of the test error of the different methods under study. For Lasso, ridge and adversarial training, we use the best $\delta$ or $\lambda$ available for each method (obtained via grid search). We note that while for $p = 1000$ optimally tuned Lasso and $\ell_\infty$-adversarial training significantly outperform the corresponding minimum $\ell_1$-norm interpolator. As $p$ increases, the performance of the three different methods becomes quite similar. It is also an example where the minimum $\ell_1$-norm outperforms the minimum $\ell_2$-norm interpolator (for large $p$). In the same setup, Figure S.12

study a choice of adversarial radius inspired by Theorem 2. We use $\delta \propto \|X\xi\|_\infty / \|\xi\|_1$ for $\xi$ a vector with zero-mean normal entries. We use a random $\xi$, since we do not know the true additive noise. Even with this approximation, $\ell_\infty$-adversarial training performs comparably with Lasso with the regularization parameter set using 5-fold cross-validation doing a full search in the hyperparameter space. The figure also provides a comparison with square-root Lasso under a similar setting.

## 8 Results for general loss functions

The following theorem can be used to generalize Proposition 1.

**Theorem 4.** Let $\mathcal{L} : \mathbb{R} \to \mathbb{R}$ be a convex and lower-semicontinuous function, for every $\delta \geq 0$,

$$\max_{\|\Delta x\| \leq \delta} \mathcal{L}\left((x + \Delta x)^\top \beta\right) = \max_{s \in \{-1,1\}} \mathcal{L}\left(x^\top \beta + \delta s \|\beta\|_*\right). \tag{12}$$

We can find a closed-formula expression for $s_* = \arg\max_{s \in \{-1,1\}} \mathcal{L}(\beta^\top x + \delta s \|\beta\|_*)$ in many cases of interest. For **regression problems**, given any non-decreasing function $\ell : \mathbb{R}^+ \to \mathbb{R}^+$, and $\mathcal{L}(x^\top \beta) = \ell(|x^\top \beta - y|)$. If $\ell$ is lower semicontinuous and convex then so will $\mathcal{L}$ and the result of the theorem holds (the squared loss $\ell(z) = z^2$ is a special case). Then $s_* = -\text{sign}(y - x^\top \beta)$ and

$$\max_{\|\Delta x\| \leq \delta} \ell(|(x + \Delta x)^\top \beta - y|) = \ell(|y - x^\top \beta| + \delta \|\beta\|_*).$$

For **classification**, let $y \in \{-1, 1\}$ and $\mathcal{L}(x^\top \beta) = \ell(y(x^\top \beta))$ where $\ell$ is a non-increasing function. If $\ell$ is lower semicontinuous and convex then so will $\mathcal{L}$ and the result of the theorem holds. Then $s_* = -y$ and we obtain:

$$\max_{\|\Delta x\| \leq \delta} \ell(y((x + \Delta x)^\top \beta)) = \ell(y(x^\top \beta) - \delta \|\beta\|_*).$$

The above result can also be applied to unsupervised learning. In the Appendix, we illustrate how Theorem 4 can be used for **dimensionality reduction**. We consider the problem of finding $P$ that minimizes the reconstruction error $\|x + PP^\top x\|_2^2$ (that yields PCA algorithm) and derive an adversarial version of it.

*Remark* 6. Over this paper we consider $x, \beta \in \mathbb{R}^p$, but Theorem 4 holds generaly for $x \in \mathcal{X}$ a vector in a Banach space endowed with norm $\| \cdot \|$ and $\beta \in \mathcal{X}^*$ a continuous linear map $\beta : \mathcal{X} \to \mathbb{R}$. Just define $x^\top \beta := \beta(x)$ and take $\| \cdot \|_*$ to be the norm of the dual space $\mathcal{X}^*$.

## 9 Conclusion

We study adversarial training in linear regression. Adversarial training allows us to depart from the traditional regularization setting where we can decompose the loss and the regularization terms, minimizing $\frac{1}{n} \sum_{i=0}^{n} \ell(x_i^\top \beta, y_i) + \Omega(\beta)$ for some penalization function $\Omega$. We show how it provides new insights into the minimum-norm interpolator, by proving a new equivalence between the two methods (Section 4). While adversarially-trained linear regression arrives at similar solutions to parameter shrinking methods in some scenarios (Section 5), it also has some advantages, for instance, the adversarial radius might be set without knowing the noise variance (Section 6). Unlike, squared-root Lasso it achieves this while still minimizing a sum of squared terms. We believe a natural next step is to provide a tailored solver that can allow for efficient solutions for models with many features, rendering adversarially-trained linear regression useful, for instance, in modeling genetics data (see one minimal example of phenotype prediction from the genotype in Section 7).

Another interesting direction is generalizing our results to classification and nonlinear models. Indeed, adversarial training is very often used in the context of neural networks and it is natural to be interested in the behavior of this class of models. Our work allows for the analysis of simplified theoretical models commonly used to study neural networks. Random Fourier feature models can be analyzed using our theory and are they studied in the numerical examples. These models can be seen as a simplified neural network model (one-layer neural network where only the last layer weights are adjusted). In Appendix A.4, we provide adversarial training after linear projections, and could, for instance, be used to analyze deep linear networks, as the ones studied in [45], [46]. Finally, Section 8 provides a result for infinite spaces, and one could attempt to use it to analyze kernel methods and infinitely-wide neural networks [47], [48]. Overall we believe that our results provide an interesting set of tools also for the analysis of nonlinear models, and could provide insight into the empirically observed interplay (see [49]) between robustness and regularization in deep neural networks.

# Acknowledgments

The authors would like to thank Carl Nettelblad for very fruitful discussions throughout the work with this research. We also thank Daniel Gedon and Dominik Baumann for their feedback on the first version of the manuscript.

TBS and DZ are financially supported by the Swedish Research Council, with the projects *Deep probabilistic regression – new models and learning algorithms* (contract number: 2021-04301) and *Counterfactual Prediction Methods for Heterogeneous Populations* (contract number:2018- 05040) and by the *Wallenberg AI, Autonomous Systems and Software Program (WASP)* funded by Knut and Alice Wallenberg Foundation. TBS, AHR and DZ by the *Kjell och Märta Beijer Foundation*. FB by the Agence Nationale de la Recher-che as part of the "Investissements d'avenir" program, reference ANR-19-P3IA0001 (PRAIRIE 3IA Institute); and by the European Research Council (grant SEQUOIA 724063).

The research was partially conducted during AHR's research visit to INRIA. The visit was financially supported by the Institute Français de Suède through the SFVE-A mobility program; and, by the European Network of AI Excellence Centres through ELISE mobility program.

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
