# Appendix

## Table of Contents

## A  Adversarial training in the overparametrized regime

### A.1  Additional proofs

The next lemma was used in the proof of Theorem 1. It formalizes the equivalence between the minimum-norm solution of $\boldsymbol{y} = \boldsymbol{X}\boldsymbol{\beta}$ and the dual problem we need to solve to obtain $\widehat{\boldsymbol{\alpha}}$.

**Lemma 1.** The following equivalence hold

$$\min_{y=\boldsymbol{X}\boldsymbol{\beta}} \|\boldsymbol{\beta}\|_* = \max_{\|\boldsymbol{\alpha}^\top \boldsymbol{X}\| \leq 1} \boldsymbol{\alpha}^\top \boldsymbol{y}. \tag{S.1}$$

Furthermore, $\widehat{\boldsymbol{\beta}}$ and $\widehat{\boldsymbol{\alpha}}$ be the arguments at optimality if and only if $\widehat{\boldsymbol{\alpha}}^\top \boldsymbol{X} \in \partial\|\widehat{\boldsymbol{\beta}}\|_*$.

*Proof of Lemma 1.*  By strong duality:

$$\min_{\boldsymbol{y}=\boldsymbol{X}\theta} \|\theta\|_* = \max_\alpha \min_\theta (\|\theta\|_* + \alpha^T(y - \boldsymbol{X}\theta)) = \max_\alpha \left( \alpha^T \boldsymbol{y} + \min_\theta (\|\theta\|_* - \alpha^T \boldsymbol{X}\theta) \right)$$

Now, the Fenchel conjugate of $\|\cdot\|_*$ is the indicator function on the ball of the norm $\|\cdot\|$. Hence, we obtain the result.  □

## A.2 Results on the robustness gap and adversarial training

Next we provide a proof of Proposition 2.

*Proof of Proposition 2.* Using Theorem 4, for any distribution on $(\boldsymbol{x}, y)$

$$\mathbb{E}_{\boldsymbol{x},y} \left[ \max_{\|\boldsymbol{\Delta x}\| \leq \delta} (y - (\boldsymbol{x} + \boldsymbol{\Delta x})^\top \boldsymbol{\beta}|^2) \right] = \mathbb{E}_{\boldsymbol{x},y} \left[ (|y - \boldsymbol{x}^\top \boldsymbol{\beta}| + \delta \|\boldsymbol{\beta}\|_*)^2 \right]$$

$$= \mathbb{E}_{\boldsymbol{x},y} \left[ (y - \boldsymbol{x}^\top \boldsymbol{\beta})^2 \right] + 2\delta \|\boldsymbol{\beta}\|_* \mathbb{E}_{\boldsymbol{x},y} \left[ |y - \boldsymbol{x}^\top \boldsymbol{\beta}| \right] + \delta^2 \|\boldsymbol{\beta}\|_*^2,$$

Now, since $0 \leq \mathbb{E}_{\boldsymbol{x},y} \left[ |y_i - \boldsymbol{x}_i^\top \boldsymbol{\beta}| \right] \leq \sqrt{\mathbb{E}_{\boldsymbol{x},y} \left[ |y_i - \boldsymbol{x}_i^\top \boldsymbol{\beta}|^2 \right]}$ (by Jensen inequality), we have that

$$\mathbb{E}_{\boldsymbol{x},y} \left[ (y - \boldsymbol{x}^\top \boldsymbol{\beta})^2 \right] + \delta^2 \|\boldsymbol{\beta}\|_*^2 \leq \mathbb{E}_{\boldsymbol{x},y} \left[ (|y - \boldsymbol{x}^\top \boldsymbol{\beta}| + \delta \|\boldsymbol{\beta}\|_*)^2 \right] \leq \left( \sqrt{\mathbb{E}_{\boldsymbol{x},y} \left[ (y_i - \boldsymbol{x}_i^\top \boldsymbol{\beta})^2 \right]} + \delta \|\boldsymbol{\beta}\|_* \right)^2.$$

If we denote, $R_*^{\mathrm{adv}}(\boldsymbol{\beta}; \delta_{\mathrm{test}}, \|\cdot\|) = \mathbb{E}_{y,\boldsymbol{x}} \left[ \max_{\|\boldsymbol{\Delta x}_i\| \leq \delta_{\mathrm{test}}} (y - (\boldsymbol{x} + \boldsymbol{\Delta x})^\top \boldsymbol{\beta})^2 \right]$ i.e., the expected adversarial squared error and $R_*(\boldsymbol{\beta}) = (y_0 - (\boldsymbol{x}_0 + \boldsymbol{\Delta x}_0)^\top \boldsymbol{\beta})^2$ the expected squared error in the absence of an adversary on a new test point. We can rewrite it as:

$$R_*(\boldsymbol{\beta}) + \delta^2 \|\boldsymbol{\beta}\|_*^2 \leq R_*^{\mathrm{adv}}(\boldsymbol{\beta}; \delta, \|\cdot\|) \leq \left( \sqrt{R_*(\boldsymbol{\beta})} + \delta \|\boldsymbol{\beta}\|_* \right)^2.$$

Rearranging:

$$\sqrt{R_*^{\mathrm{adv}}(\boldsymbol{\beta}; \delta, \|\cdot\|)} - \sqrt{R_*(\boldsymbol{\beta})} \leq \delta \|\boldsymbol{\beta}\|_* \leq \sqrt{R_*^{\mathrm{adv}}(\boldsymbol{\beta}; \delta, \|\cdot\|) - R_*(\boldsymbol{\beta})}. \tag{S.2}$$

Now for the empirical adversarial distribution:

$$R^{\mathrm{adv}}(\widehat{\boldsymbol{\beta}}; \bar{\delta}, \|\cdot\|) = \frac{1}{n} \sum_{i=1}^n (|y_i - \boldsymbol{x}_i^\top \widehat{\boldsymbol{\beta}}| + \bar{\delta} \|\boldsymbol{\beta}\|_*)^2$$

$$= \bar{\delta}^2 \|\boldsymbol{\beta}\|_*^2$$

where the last step follows from considering that $\widehat{\boldsymbol{\beta}}$ is an interpolator (hence $y_i - \boldsymbol{x}_i^\top \widehat{\boldsymbol{\beta}} = 0, \forall i$). Plugging this result back into (S.2). Let $\delta = \delta_{\mathrm{test}}$ and $\delta' = \delta_{\mathrm{train}}$ $\qquad \square$

The next result holds for mismatched $\ell_\infty$ and $\ell_2$-adversarial attacks

---

**Proposition 7.** Let $\widehat{\boldsymbol{\beta}}$ be the minimum $\|\cdot\|_*$-norm interpolator, than

$$\sqrt{R_*^{\mathrm{adv}}(\boldsymbol{\beta}; \delta_{\mathrm{test}}, \|\cdot\|_\infty)} - \sqrt{R_*(\boldsymbol{\beta})} \leq \sqrt{p} \frac{\delta_{\mathrm{test}}}{\bar{\delta}} \sqrt{R^{\mathrm{adv}}(\boldsymbol{\beta}; \bar{\delta}, \|\cdot\|_2)} \tag{S.3}$$

---

*Proof.* A similar derivation as in the proof of Proposition 2 yields:

$$\sqrt{R_*^{\mathrm{adv}}(\widehat{\boldsymbol{\beta}}; \delta, \|\cdot\|_\infty)} - \sqrt{R_*(\boldsymbol{\beta})} \leq \delta \|\widehat{\boldsymbol{\beta}}\|_1 \leq \sqrt{R_*^{\mathrm{adv}}(\widehat{\boldsymbol{\beta}}; \delta, \|\cdot\|_\infty) - R_*(\widehat{\boldsymbol{\beta}})}$$

$$R^{\mathrm{adv}}(\widehat{\boldsymbol{\beta}}; \bar{\delta}, \|\cdot\|_2) = \bar{\delta}^2 \|\widehat{\boldsymbol{\beta}}\|_2^2.$$

Now since $\|\widehat{\boldsymbol{\beta}}\|_2 \leq \|\widehat{\boldsymbol{\beta}}\|_1 \leq \sqrt{p} \|\widehat{\boldsymbol{\beta}}\|_2$ the result follows for $\delta = \delta_{\mathrm{test}}$. $\qquad \square$

## A.3 Bounds on $\bar{\delta}$

We point that although the exact value of $\bar{\delta}$ requires the solution of the dual problem. It is easy to come up with upper and lower bounds that depend only on $\boldsymbol{X}$. On the one hand, by construction $\|\boldsymbol{X}^\top \widehat{\boldsymbol{\alpha}}\| \leq 1$, hence,

$$\frac{1}{\|\widehat{\boldsymbol{\alpha}}\|_\infty} \geq \frac{\|\boldsymbol{X}^\top \widehat{\boldsymbol{\alpha}}\|}{\|\widehat{\boldsymbol{\alpha}}\|_\infty}.$$

On the other hand, $\boldsymbol{X}^\top \widehat{\boldsymbol{\alpha}} \in \partial\|\widehat{\boldsymbol{\beta}}\|_*$ such that $\widehat{\boldsymbol{\alpha}}^\top \boldsymbol{X}\widehat{\boldsymbol{\beta}} = \|\widehat{\boldsymbol{\beta}}\|_*$. Hence a simple application of Hölder inequality yields:

$$\frac{\|\boldsymbol{X}\widehat{\boldsymbol{\beta}}\|_1}{\|\widehat{\boldsymbol{\beta}}\|_*} \geq \frac{1}{\|\widehat{\boldsymbol{\alpha}}\|_\infty}$$

Now since, $n\bar{\delta} = \frac{1}{\|\widehat{\boldsymbol{\alpha}}\|_\infty}$ we have that:

$$\inf_{\boldsymbol{u}\in\mathbb{R}^n} \frac{\|\boldsymbol{X}^\top \boldsymbol{u}\|}{\|\boldsymbol{u}\|_\infty} \leq n\bar{\delta} \leq \sup_{\boldsymbol{v}\in\mathbb{R}^m} \frac{\|\boldsymbol{X}\boldsymbol{v}\|_1}{\|\boldsymbol{v}\|_*}$$

For instance, for $\ell_\infty$-adversarial attacks, this speciallize to

$$\inf_{\boldsymbol{u}\in\mathbb{R}^n} \frac{\|\boldsymbol{X}^\top \boldsymbol{u}\|_\infty}{\|\boldsymbol{u}\|_\infty} \leq n\bar{\delta} \leq \sup_{\boldsymbol{v}\in\mathbb{R}^m} \frac{\|\boldsymbol{X}\boldsymbol{v}\|_1}{\|\boldsymbol{v}\|_1}$$

Let, $\sigma_1(\boldsymbol{X}) \geq \cdots \geq \sigma_n(\boldsymbol{X})$ singular values of the matrix $\boldsymbol{X} \in \mathbb{R}^{n\times m}$. We have that $\sigma_1(\boldsymbol{X}) = \sup_{\boldsymbol{v}\in\mathbb{R}^m} \frac{\|\boldsymbol{X}\boldsymbol{v}\|_2}{\|\boldsymbol{v}\|_2}$ and $\sigma_n(\boldsymbol{X}) = \inf_{\boldsymbol{u}\in\mathbb{R}^n} \frac{\|\boldsymbol{X}^\top \boldsymbol{u}\|_2}{\|\boldsymbol{u}\|_2}$ Now we can use standard norm inequalities, let $\boldsymbol{w} \in \mathbb{R}^m$: $\|\boldsymbol{w}\|_2 \leq \|\boldsymbol{w}\|_1 \leq \sqrt{m}\|\boldsymbol{w}\|_2$ and $\|\boldsymbol{w}\|_\infty \leq \|\boldsymbol{w}\|_2 \leq \sqrt{m}\|\boldsymbol{w}\|_\infty$ to obtain, that for for $\ell_\infty$-adversarial attacks:

$$\frac{1}{\sqrt{m}}\sigma_n(\boldsymbol{X}) \leq n\bar{\delta} \leq \sqrt{m}\sigma_1(\boldsymbol{X})$$

Similarly $\ell_2$-adversarial attacks one can show that $\sigma_n \leq n\bar{\delta} \leq \sqrt{m}\sigma_1$.

It is also possible to establish a relationship with the minimum-norm interpolator. Let $\widehat{\boldsymbol{\beta}}$ be the minimum norm interpolators, using Hölder inequality

$$\|\widehat{\boldsymbol{\beta}}\|_* = \widehat{\boldsymbol{\alpha}}^\top \boldsymbol{y} \leq \|\widehat{\boldsymbol{\alpha}}\|_\infty \|\boldsymbol{y}\|_1$$

And it follows that :

$$\bar{\delta}\|\widehat{\boldsymbol{\beta}}\|_* \leq \frac{1}{n}\|\boldsymbol{y}\|_1$$

Now, if we assume that the data was generated as $y_i = \boldsymbol{x}_i^\top \boldsymbol{\beta}^* + \varepsilon_i$ for $i = 1, \cdots, n$. From Hölder inequality and the definition of the dual problem

$$\widehat{\boldsymbol{\alpha}}^\top \boldsymbol{X}\boldsymbol{\beta}^* \leq \|\widehat{\boldsymbol{\alpha}}^\top \boldsymbol{X}\|\|\boldsymbol{\beta}^*\|_* \leq \|\boldsymbol{\beta}^*\|_* \tag{S.4}$$

Now:

$$\begin{aligned}
\|\widehat{\boldsymbol{\beta}}\|_* &\overset{(a)}{=} \widehat{\boldsymbol{\alpha}}^\top \boldsymbol{y} \overset{(b)}{=} \widehat{\boldsymbol{\alpha}}^\top \boldsymbol{X}\boldsymbol{\beta}^* + \widehat{\boldsymbol{\alpha}}^\top \boldsymbol{\varepsilon} \\
&\overset{(c)}{\leq} \|\boldsymbol{\beta}^*\|_* + \widehat{\boldsymbol{\alpha}}^\top \boldsymbol{\varepsilon} \\
&\leq \|\boldsymbol{\beta}^*\|_* + \|\widehat{\boldsymbol{\alpha}}\|_\infty \|\boldsymbol{\varepsilon}\|_1 \\
&= \|\boldsymbol{\beta}^*\|_* + \frac{1}{n}\frac{\|\boldsymbol{\varepsilon}\|_1}{\bar{\delta}}
\end{aligned}$$

where (a) follows Lemma 1, (b) follows from the data model definition and finaly (c) follows from Eq. (S.4).

## A.4 Linear maps and random projections

As an extension, we consider the following framework: We consider a matrix $\boldsymbol{S} \in \mathbb{R}^{p\times d}$ that maps from the input space $\mathbb{R}^d$ to a feature space $\mathbb{R}^p$. For instance, in Bach [40] this scenario—with the entries of $\boldsymbol{S}$ sampled from a Rademacher distribution—is used to study the phenomena of double-descent. For this case, the adversarial problem consists of finding a parameter $\widehat{\boldsymbol{\beta}}$ that minimizes:

$$R_{\boldsymbol{S}}^{\mathrm{adv}}(\boldsymbol{\beta}; \delta, \|\cdot\|) = \frac{1}{n}\sum_{i=1}^n \max_{\|\boldsymbol{\Delta x}_i\|\leq\delta} |y_i - (\boldsymbol{x}_i + \boldsymbol{\Delta x}_i)^\top \boldsymbol{S}^\top \boldsymbol{\beta}|^2. \tag{S.5}$$

Equation (S.5) also allow for reformulation:

**Proposition 8** (Dual formulation: linear maps). Let $\| \cdot \|_*$ be the dual norm of $\| \cdot \|$, then

$$R_{\boldsymbol{S}}^{\mathrm{adv}}(\boldsymbol{\beta}; \delta, \| \cdot \|) = \frac{1}{n} \sum_{i=1}^{n} \left( |y_i - \boldsymbol{x}_i^\top \boldsymbol{S}^\top \boldsymbol{\beta}| + \delta \|\boldsymbol{S}^\top \boldsymbol{\beta}\|_* \right)^2. \tag{S.6}$$

this is a direct consequence of the more general result Theorem 7. We also have the following theorem relating the adversarial training solution to the minimum $\| \cdot \|_*$-norm interpolator.

**Theorem 5.** Assume the matrix $\boldsymbol{X}\boldsymbol{S}^\top \in \mathbb{R}^{n \times p}$ has full row rank (i.e., $\mathrm{rank}(\boldsymbol{X}\boldsymbol{S}^\top) = n$). The minimum-norm solution

$$\widehat{\boldsymbol{\beta}} = \arg\min_{\boldsymbol{\beta}} \|\boldsymbol{S}^\top \boldsymbol{\beta}\|_* \quad \text{subject to} \quad \boldsymbol{X}\boldsymbol{S}^\top \boldsymbol{\beta} = \boldsymbol{y}, \tag{S.7}$$

minimizes the adversarial risk $R_{\boldsymbol{S}}^{\mathrm{adv}}(\theta, \delta, \| \cdot \|)$ if and only if $\delta \in (0, \bar{\delta}]$. For $\bar{\delta} = \frac{1}{n\|\widehat{\boldsymbol{\alpha}}\|_\infty}$ where $\widehat{\boldsymbol{\alpha}}$ denote the solution of the dual problem $\max_{\|\boldsymbol{\alpha}^\top \boldsymbol{X}\boldsymbol{P}\| \leq 1} \boldsymbol{\alpha}^\top y$, where $\boldsymbol{P} = \boldsymbol{S}^\top (\boldsymbol{S}\boldsymbol{S}^\top)^{-1}\boldsymbol{S}$.

# B  Adversarial training and parameter shrinking methods

## B.1  Background

The subderivative of a function $\omega : \mathbb{R}^p \to \mathbb{R}$ evaluated at a point $\boldsymbol{\beta}_0$ is the set

$$\partial\omega(\boldsymbol{\beta}_0) = \{\boldsymbol{v} \in \mathbb{R}^p : \omega(\boldsymbol{\beta}) - \omega(\boldsymbol{\beta}_0) \geq \boldsymbol{v}(\boldsymbol{\beta} - \boldsymbol{\beta}_0) \; \forall \boldsymbol{\beta} \in \mathbb{R}^p\}.$$

In this section, for convenience, we will drop the two last arguments of $R^{\mathrm{adv}}(\boldsymbol{\beta}; \delta, \| \cdot \|)$ and denote it only by $R^{\mathrm{adv}}(\boldsymbol{\beta})$. Let $\mathcal{L}_i(\boldsymbol{\beta}) = |y_i - \boldsymbol{x}_i^\top \boldsymbol{\beta}| + \delta\|\boldsymbol{\beta}\|_*$, then the partial derivative of $R^{\mathrm{adv}}$ with respect to $\boldsymbol{\beta}$ is

$$\partial R^{\mathrm{adv}}(\boldsymbol{\beta}) = \frac{2}{n} \sum_{i=1}^{n} \mathcal{L}_i(\boldsymbol{\beta})\partial\mathcal{L}_i(\boldsymbol{\beta}), \text{ where } \partial L_i(\boldsymbol{\beta}) = \boldsymbol{x}_i\partial|\boldsymbol{x}_i^\top \boldsymbol{\beta} - y_i| + \delta\partial\|\boldsymbol{\beta}\|_*, \tag{S.8}$$

where

$$\partial|a| = \begin{cases} \{1\} \text{ if } a > 0 \\ \{-1\} \text{ if } a < 0 \\ \{\gamma : \gamma \in [-1, 1]\} \text{ if } a = 0, \end{cases}$$

and:

$$\partial\|\boldsymbol{\beta}\|_* = \left\{\boldsymbol{\alpha} : \|\boldsymbol{\alpha}\| \leq 1, \boldsymbol{\alpha}^\top\boldsymbol{\beta} = \|\boldsymbol{\beta}\|_*\right\}. \tag{S.9}$$

Since, $R^{\mathrm{adv}}(\boldsymbol{\beta})$ is a convex function of $\boldsymbol{\beta}$, we have that $\widehat{\boldsymbol{\beta}}$ is a solution of $\min R^{\mathrm{adv}}(\boldsymbol{\beta})$ iff $\boldsymbol{0} \in \partial R^{\mathrm{adv}}(\widehat{\boldsymbol{\beta}})$.

## B.2  Zero solution to adversarial training: Proposition 3

Proposition 3 stated that *The zero solution $\widehat{\boldsymbol{\beta}} = \boldsymbol{0}$ minimizes the adversarial training iff $\delta \geq \frac{\|\boldsymbol{X}^\top\boldsymbol{y}\|}{\|\boldsymbol{y}\|_1}$.* Indeed, one can notice from Figure 3 and Figures S.2 and S.3 there is a threshold of $\delta$, such that for all $\delta$ above such threshold the solution is identical to zero. We provide proof for this theorem next.

*Proof of Proposition 3.* On the one hand,

$$\partial R^{\mathrm{adv}}(\boldsymbol{0}) = \frac{2}{n} \sum_{i=1}^{n} |y_i|(\boldsymbol{x}_i\partial|y_i| + \delta\partial\|\boldsymbol{0}\|_*).$$

In matrix form:

$$\partial R^{\mathrm{adv}}(\boldsymbol{0}) = \frac{2}{n} \left( \boldsymbol{X}^\top\boldsymbol{y} + \delta\|\boldsymbol{y}\|_1\partial\|\boldsymbol{0}\|_* \right)$$

where we used that $|y_i|\partial|y_i| = y_i$. Now, $\partial\|\boldsymbol{0}\|_* = \{\boldsymbol{\alpha} : \|\boldsymbol{\alpha}\| \leq 1\}$, hence for $\boldsymbol{z} \in \partial\|\boldsymbol{0}\|_*$, we obtain from triangular inequality that:

$$\|\boldsymbol{X}^\top\boldsymbol{y}\| - \delta\|\boldsymbol{y}\|_1 \leq \left\|\boldsymbol{X}^\top\boldsymbol{y} + \delta\|\boldsymbol{y}\|_1\boldsymbol{z}\right\|.$$

On the one hand, if $\frac{\|\boldsymbol{X}^\top\boldsymbol{y}\|}{\|\boldsymbol{y}\|_1} > \delta$, the left-hand-side of the above inequality is larger than zero, and $\boldsymbol{0} \notin \partial R^{\mathrm{adv}}(\boldsymbol{0})$. On the other hand, if $\frac{\|\boldsymbol{X}^\top\boldsymbol{y}\|}{\|\boldsymbol{y}\|_1} \leq \delta$, we have $\boldsymbol{z} = \boldsymbol{X}^T\boldsymbol{y}/(\delta\|\boldsymbol{y}\|_1) \in \partial\|\boldsymbol{0}\|_*$ (since $\|\boldsymbol{z}\| \leq 1$), and:

$$\boldsymbol{X}^\top\boldsymbol{y} + \delta\|\boldsymbol{y}\|_1\boldsymbol{z} = \boldsymbol{0}.$$

Therefore $\boldsymbol{0} \in \partial R^{\mathrm{adv}}(\boldsymbol{0})$ and the proof is complete. $\qquad\square$

## B.3  On the similarities of regularization paths: Proposition 4 and extensions

The next phenomenon we want to explain is the similarity between Lasso and $\ell_\infty$-adversarial attacks regularization paths. As well as the similarities between ridge and $\ell_2$-adversarial attacks regularization paths.

> **Theorem 6.** Let $\widehat{\boldsymbol{\beta}}(\delta)$ be the minimizer of $R^{\mathrm{adv}}(\boldsymbol{\beta}; \delta, \|\cdot\|)$, define the vector $\boldsymbol{s}(\delta) = \mathrm{sign}(\boldsymbol{y} - \boldsymbol{X}\widehat{\boldsymbol{\beta}}(\delta))$. Assume that $y_i \neq \boldsymbol{x}_i^\top\widehat{\boldsymbol{\beta}}(\delta)$ for every $i$. If $\boldsymbol{X}^\top\boldsymbol{s}(\delta) = \boldsymbol{0}$ than $\widehat{\boldsymbol{\beta}}(\delta)$ is a minimizer of
>
> $$\frac{1}{n}\|\boldsymbol{X}\boldsymbol{\beta} - \boldsymbol{y}\|_2^2 + \left(\delta\|\boldsymbol{\beta}\|_* + \frac{1}{n}\boldsymbol{s}(\delta)^\top\boldsymbol{y}\right)^2, \qquad (S.10)$$

*Proof.* We will prove first that if $\boldsymbol{X}^\top\boldsymbol{s}(\delta) = 0$, than a minimizer of $R^{\mathrm{adv}}(\boldsymbol{\beta}; \delta, \|\cdot\|)$ is also a minimizer of:

$$\frac{1}{n}\|\boldsymbol{X}\boldsymbol{\beta} - \boldsymbol{y}\|_2^2 + \left(\delta\|\boldsymbol{\beta}\|_* + \frac{1}{n}\boldsymbol{s}(\delta)^\top\boldsymbol{y}\right)^2 \qquad (S.11)$$

is also a minimizer of $R^{\mathrm{adv}}(\boldsymbol{\beta}; \delta, \|\cdot\|)$. Multiplying the terms in (S.8) and putting into matrix form:

$$\frac{1}{2}\partial R^{\mathrm{adv}}(\boldsymbol{\beta}) = \frac{1}{n}\boldsymbol{X}^\top(\boldsymbol{X}\boldsymbol{\beta} - \boldsymbol{y}) + \frac{\delta\|\boldsymbol{\beta}\|_*}{n}\boldsymbol{X}^\top\partial\|\boldsymbol{X}\boldsymbol{\beta} - \boldsymbol{y}\|_1 + \left(\frac{1}{n}\|\boldsymbol{X}\boldsymbol{\beta} - \boldsymbol{y}\|_1 + \delta\|\boldsymbol{\beta}\|_*\right)\delta\partial\|\boldsymbol{\beta}\|_*$$

Now, we have that $\partial\|\boldsymbol{X}\widehat{\boldsymbol{\beta}} - \boldsymbol{y}\|_1 = \{-\boldsymbol{s}(\delta)\}$ and that $\|\boldsymbol{X}\widehat{\boldsymbol{\beta}} - \boldsymbol{y}\|_1 = \boldsymbol{s}(\delta)^\top(\boldsymbol{y} - \boldsymbol{X}\widehat{\boldsymbol{\beta}}) = \boldsymbol{s}(\delta)^\top\boldsymbol{y}$, hence:

$$\frac{1}{2}\partial R^{\mathrm{adv}}(\widehat{\boldsymbol{\beta}}) = \frac{1}{n}\boldsymbol{X}^\top(\boldsymbol{X}\widehat{\boldsymbol{\beta}} - \boldsymbol{y}) + \left(\delta\|\widehat{\boldsymbol{\beta}}\|_* + \frac{1}{n}\boldsymbol{s}(\delta)^\top\boldsymbol{y}\right)\delta\partial\|\widehat{\boldsymbol{\beta}}\|_*$$

The left-hand side is the subderivative of (S.11) evaluated at $\widehat{\boldsymbol{\beta}}$. Since $\widehat{\boldsymbol{\beta}}$ is the minimizer of $R^{\mathrm{adv}}(\widehat{\boldsymbol{\beta}})$, than $\boldsymbol{0} \in \partial R^{\mathrm{adv}}(\widehat{\boldsymbol{\beta}})$ and it follows that $\widehat{\boldsymbol{\beta}}$ is also a minimizer of (S.11). $\qquad\square$

Proposition 4 is weaker than necessary. But we choose it to be part of the main text instead instead of Theorem 6 because it has a cleaner interpretation: it only depends on the norm of $\widehat{\boldsymbol{\beta}}$ and not on its direction. And $\boldsymbol{X}^\top\boldsymbol{1} = 0$ has the interpretation that the data has been normalized. The proof follows directly from the Theorem.

*Proof of Proposition 4.* We can use the Hölder inequality $|\boldsymbol{x}_i^\top\widehat{\boldsymbol{\beta}}| \leq \|\widehat{\boldsymbol{\beta}}\|_*\|\boldsymbol{x}_i\|$, to show that if $\|\widehat{\boldsymbol{\beta}}\|_*\|\boldsymbol{x}_i\| \leq |y_i|$ than $|\boldsymbol{x}_i^\top\widehat{\boldsymbol{\beta}}(\delta)| \leq |y_i|$ for all $i$. In this case, $\boldsymbol{s}(\delta) = \mathrm{sign}(\boldsymbol{y})$. If additionally if $\boldsymbol{y} > 0$, the Theorem implies that as long as $\boldsymbol{X}^\top\boldsymbol{1} = 0$ then $\widehat{\boldsymbol{\beta}}(\delta)$ is the minimizer of:

$$\frac{1}{n}\|\boldsymbol{X}\boldsymbol{\beta} - \boldsymbol{y}\|_2^2 + \left(\delta\|\boldsymbol{\beta}\|_* + \frac{1}{n}\|\boldsymbol{y}\|_1\right)^2.$$

$\qquad\square$

We notice that the theorem conclusion holds even under the (less strict) condition, $|\boldsymbol{x}_i^\top\widehat{\boldsymbol{\beta}}(\delta)| \leq |y_i|$ for every $i$. Figure 1 highlights the part of the regularization path for which this condition holds. Showing that even for values of $\delta$ for which this condition does not hold, the regularization paths are still extremely similar.

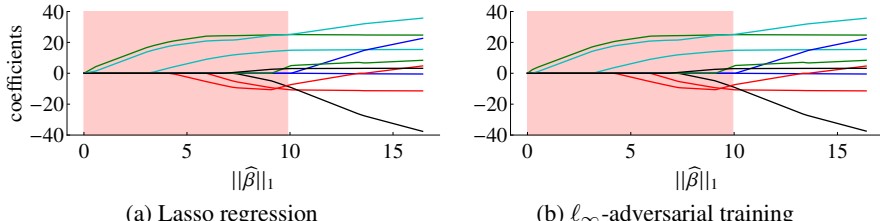

(a) Lasso regression   (b) $\ell_\infty$-adversarial training

Figure S.1: **Regularization paths: diabetes dataset**. On the horizontal axis, we give the $\|\widehat{\boldsymbol{\beta}}\|_1$. On the vertical axis, we show the coefficients of the learned linear model. The hashed area gives the values of $\widehat{\boldsymbol{\beta}}$ for which $|\boldsymbol{x}_i^\top \widehat{\boldsymbol{\beta}}(\delta)| \leq |y_i|$ for every $i$.

This can be explained from the same argument use in Theorem 6. Assume the hypothesis that the covariates $\{\boldsymbol{x}_i\}_{i=1}^n$ are i.i.d. and are sampled from a symmetric and zero-mean distribution, i.e. $\boldsymbol{x} \sim -\boldsymbol{x}$ and $\mathbb{E}[\boldsymbol{x}] = 0$. In this case, we notice that if $s \in \{-1, 1\}$ then by symmetry of the distribution $s\boldsymbol{x} \sim \boldsymbol{x}$, and we have that $\mathbb{E}[s\boldsymbol{x}] = \mathbf{0}$. Now, from the law of large numbers, $\frac{1}{n}\sum s_i \boldsymbol{x}_i \to 0$ as $n \to 0$. Let $\widehat{\boldsymbol{\beta}}$ be the solution of

$$\frac{1}{n}\|\boldsymbol{X}\boldsymbol{\beta} - \boldsymbol{y}\|_2^2 + \left(\delta\|\boldsymbol{\beta}\|_* + \frac{1}{n}\boldsymbol{s}^\top \boldsymbol{y}\right)^2.$$

The same argument used in the proof of Theorem can than be used to show that in this case $\partial R^{\mathrm{adv}}(\widehat{\boldsymbol{\beta}}) \to 0$ as $n \to \infty$. Hence, for large enought $n$ both problems have approximately the same solution.

## B.4   Regularization path for Gaussian covariates

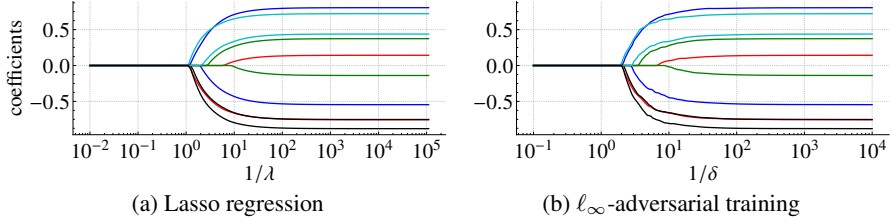

(a) Lasso regression   (b) $\ell_\infty$-adversarial training

Figure S.2: **Regularization paths: Gaussian covariates.** Lasso and $\ell_\infty$-adversarial training. On the horizontal axis, we give the inverse of the regularization parameter (in log scale). On the vertical axis we show the coefficients of the learned linear model.

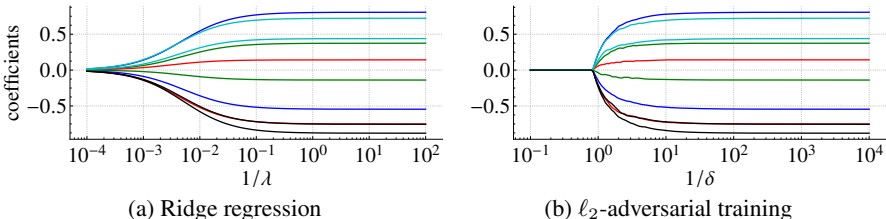

(a) Ridge regression   (b) $\ell_2$-adversarial training

Figure S.3: **Regularization paths: Gaussian covariates**. Ridge regression and $\ell_2$-adversarial training. On the horizontal axis, we give the inverse of the regularization parameter (in log scale). On the vertical axis we show the coefficients of the learned linear model.

## B.5 Additional details on Figure 1: relationship between $\|\widehat{\boldsymbol{\beta}}\|_1$ and $\lambda$ and $\delta$

Figure S.4(a) illustrate the relationship between $\|\widehat{\boldsymbol{\beta}}\|_1$ and $\lambda$ and $\delta$. Figure S.4(b-c) shows the same coefficients as in Figure 1, but considers $1/\lambda$ and $1/\delta$ is the $x$-axis instead of $\|\widehat{\boldsymbol{\beta}}\|_1$.

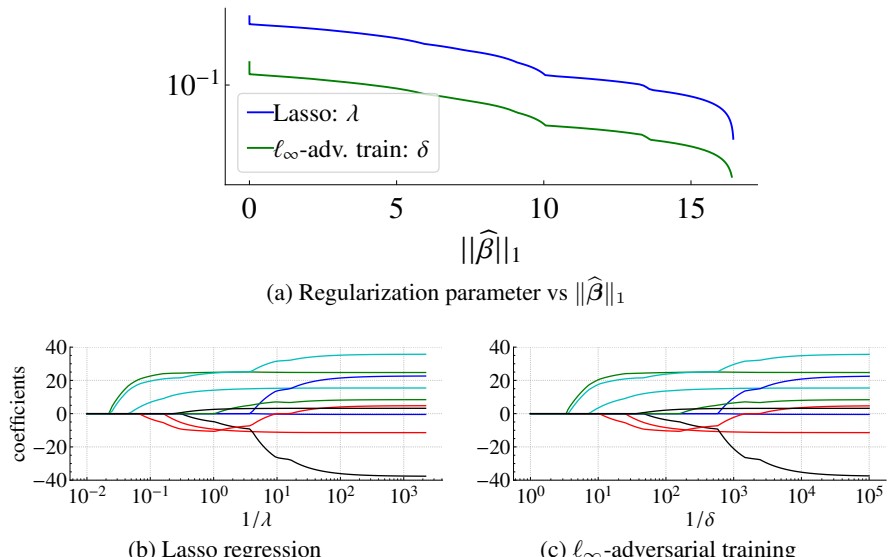

(a) Regularization parameter vs $\|\widehat{\boldsymbol{\beta}}\|_1$

(b) Lasso regression

(c) $\ell_\infty$-adversarial training

Figure S.4: **Regularization parameter vs $\|\widehat{\boldsymbol{\beta}}\|_1$**. In (a), we show the relationship between $\lambda$ and $\delta$ and $\|\widehat{\boldsymbol{\beta}}\|_1$. In (b), we show the regularization paths estimated in the Diabetes dataset [18] for Lasso with $1/\lambda$ in the $x$-axis; and, in (c) for $\ell_\infty$-adversarial attacks, with $1/\delta$ in the $x$-axis.

## C  Relation to robust regression and square-root Lasso

### C.1  Proof of Proposition 5

*Proof.* It follows from noting that

$$R_p^{\mathrm{adv}}(\boldsymbol{\beta}; \delta) = \frac{1}{n} \max_{\boldsymbol{\Delta} \in \mathcal{R}_p(\delta)} \|\boldsymbol{y} - (\boldsymbol{X} + \boldsymbol{\Delta})\boldsymbol{\beta}\|_2^2 = \frac{1}{n} \left( \max_{\boldsymbol{\Delta} \in \mathcal{R}_p(\delta)} \|\boldsymbol{y} - (\boldsymbol{X} + \boldsymbol{\Delta})\boldsymbol{\beta}\|_2 \right)^2.$$

Where the first equality follows from the definition of adversarial training and the last equality from the fact that the function $h(z) = \frac{1}{n}z^2$ is monotonically increasing for $z \geq 0$. Repeating the same argument, but now for the minimization, implies that $R_p^{\mathrm{adv}}(\boldsymbol{\beta}; \delta)$ has the same minimizer as $\max_{\boldsymbol{\Delta} \in \mathcal{R}_p} \|y - (\boldsymbol{X} + \boldsymbol{\Delta})\boldsymbol{\beta}\|_2$. $\qquad\square$

### C.2  Error bounds for $\ell_\infty$-adversarial training for sparse recovery (Proof of Theorem 2)

In this section, we will consider exclusively the empirical $\ell_\infty$-adversarial risk:

$$R^{\mathrm{adv}}(\boldsymbol{\beta}; \delta, \|\cdot\|_\infty) = \frac{1}{n} \sum_{i=1}^{n} \left( |y_i - \boldsymbol{x}_i^\top \boldsymbol{\beta}| + \delta\|\boldsymbol{\beta}\|_1 \right)^2.$$

For convenience, we will denote it only $R^{\mathrm{adv}}(\boldsymbol{\beta})$ in this section and we denote $\widehat{\boldsymbol{\beta}}$ the minimizer of this optimization problem. Moreover, we assume that the data was generated as:

$$y_i = \boldsymbol{x}_i^\top \boldsymbol{\beta}^* + \varepsilon_i, \tag{S.12}$$

where $\boldsymbol{\beta}^*$ is the parameter vector used to generate the data. Again we denote $\boldsymbol{X}$ the matrix of vectors $\boldsymbol{x}_i$ stacked and $\boldsymbol{y}$ and $\boldsymbol{\varepsilon}$ the vectors of stacked outputs and noise components respectively. In this

case, Theorem 2 states that for $\delta > \delta^* = 3\frac{\|\boldsymbol{X}^\top \boldsymbol{\varepsilon}\|_\infty}{\|\boldsymbol{\varepsilon}\|_1}$, *the prediction error of $\ell_\infty$-adversarial training satisfies the bound:*

$$\frac{1}{n}\|\boldsymbol{X}(\widehat{\boldsymbol{\beta}} - \boldsymbol{\beta}^*)\|_2^2 \leq 8\delta\|\boldsymbol{\beta}^*\|_1 \left(\frac{1}{n}\|\boldsymbol{\varepsilon}\|_1 + 10\delta\|\boldsymbol{\beta}^*\|_1\right).$$

We present the proof for this result next.

*Proof of Theorem 2.* Let $\mathcal{L}_i(\boldsymbol{\beta}) = |y_i - \boldsymbol{x}_i^\top\boldsymbol{\beta}| + \delta\|\boldsymbol{\beta}\|_1$, then

$$\partial R^{\mathrm{adv}}(\boldsymbol{\beta}) = \frac{2}{n}\sum_{i=1}^{n}\mathcal{L}_i(\boldsymbol{\beta})\partial\mathcal{L}_i(\boldsymbol{\beta}), \text{ where } \partial L_i(\boldsymbol{\beta}) = \boldsymbol{x}_i\partial|y_i - \boldsymbol{x}_i^\top\boldsymbol{\beta}| + \delta\partial\|\boldsymbol{\beta}\|_1.$$

Multiplying the terms and expanding

$$\frac{1}{2}\partial R^{\mathrm{adv}}(\boldsymbol{\beta}) = \frac{1}{n}\boldsymbol{X}^\top(\boldsymbol{X}\boldsymbol{\beta} - \boldsymbol{y}) + \frac{\delta\|\boldsymbol{\beta}\|_1}{n}\boldsymbol{X}^\top\partial\|\boldsymbol{X}\boldsymbol{\beta} - \boldsymbol{y}\|_1 + \left(\frac{1}{n}\|\boldsymbol{X}\boldsymbol{\beta} - \boldsymbol{y}\|_1 + \delta\|\boldsymbol{\beta}\|_1\right)\delta\partial\|\boldsymbol{\beta}\|_1.$$

Now, since $\boldsymbol{0} \in \partial R^{\mathrm{adv}}(\widehat{\boldsymbol{\beta}})$ we must have $\widehat{\boldsymbol{z}} \in \partial\|\boldsymbol{X}\boldsymbol{\beta} - \boldsymbol{y}\|_1$ and $\widehat{\boldsymbol{w}} \in \partial\|\boldsymbol{\beta}\|_1$ such that:

$$\boldsymbol{0} = \frac{1}{n}\boldsymbol{X}^\top(\boldsymbol{X}\widehat{\boldsymbol{\beta}} - \boldsymbol{y}) + \frac{\delta\|\widehat{\boldsymbol{\beta}}\|_1}{n}\boldsymbol{X}^\top\widehat{\boldsymbol{z}} + \delta\left(\frac{1}{n}\|\boldsymbol{X}\widehat{\boldsymbol{\beta}} - \boldsymbol{y}\|_1 + \delta\|\widehat{\boldsymbol{\beta}}\|_1\right)\widehat{\boldsymbol{w}}. \tag{S.13}$$

Let us denote,

$$\widehat{\boldsymbol{\Delta}} = \widehat{\boldsymbol{\beta}} - \boldsymbol{\beta}^*.$$

Taking the dot product of both sides of Eq (S.13) with $\widehat{\boldsymbol{\Delta}}$, making the substitution $\boldsymbol{X}\widehat{\boldsymbol{\beta}} - \boldsymbol{y} = \boldsymbol{X}\widehat{\boldsymbol{\Delta}} - \boldsymbol{\varepsilon}$ and rearranging we find that

$$\frac{1}{n}\|\boldsymbol{X}\widehat{\boldsymbol{\Delta}}\|_2^2 = \frac{1}{n}(\boldsymbol{X}\widehat{\boldsymbol{\Delta}})^\top\boldsymbol{\varepsilon} - \frac{\delta\|\widehat{\boldsymbol{\beta}}\|_1}{n}(\boldsymbol{X}\widehat{\boldsymbol{\Delta}})^\top\widehat{\boldsymbol{z}} - \delta\left(\frac{1}{n}\|\boldsymbol{X}\widehat{\boldsymbol{\Delta}} - \boldsymbol{\varepsilon}\|_1 + \delta\|\widehat{\boldsymbol{\beta}}\|_1\right)\widehat{\boldsymbol{\Delta}}^\top\widehat{\boldsymbol{w}}. \tag{S.14}$$

Next, we bound each of the terms highlighted above. From (S.9), since $\widehat{\boldsymbol{w}} \in \partial\|\widehat{\boldsymbol{\beta}}\|_1$, we have that $\widehat{\boldsymbol{w}}^\top\widehat{\boldsymbol{\beta}} = \|\widehat{\boldsymbol{\beta}}\|_1$ and $\widehat{\boldsymbol{w}}^\top\boldsymbol{\beta}^* \leq \|\widehat{\boldsymbol{w}}\|_\infty\|\boldsymbol{\beta}^*\|_1 \leq \|\boldsymbol{\beta}^*\|_1$. Therefore:

$$-\widehat{\boldsymbol{w}}^\top\widehat{\boldsymbol{\Delta}} \leq \|\boldsymbol{\beta}^*\|_1 - \|\widehat{\boldsymbol{\beta}}\|_1.$$

Similarly, since $\widehat{\boldsymbol{z}} \in \partial\|\boldsymbol{X}^\top\boldsymbol{\beta} - \boldsymbol{y}\|_1$, we have that

$$-\widehat{\boldsymbol{z}}^\top\boldsymbol{X}\widehat{\boldsymbol{\Delta}} = -\widehat{\boldsymbol{z}}^\top(\boldsymbol{X}\widehat{\boldsymbol{\Delta}} - \boldsymbol{\varepsilon}) - \widehat{\boldsymbol{z}}^\top\boldsymbol{\varepsilon} \leq -\|\boldsymbol{X}\widehat{\boldsymbol{\Delta}} - \boldsymbol{\varepsilon}\|_1 + \|\boldsymbol{\varepsilon}\|_1.$$

Moreover, we have that:

$$\frac{1}{n}(\boldsymbol{X}\widehat{\boldsymbol{\Delta}})^\top\boldsymbol{\varepsilon} \overset{(a)}{\leq} \frac{1}{n}\|\boldsymbol{X}^\top\boldsymbol{\varepsilon}\|_\infty\|\widehat{\boldsymbol{\Delta}}\|_1 \overset{(b)}{\leq} \frac{\delta}{3n}\|\boldsymbol{\varepsilon}\|_1\|\widehat{\boldsymbol{\Delta}}\|_1,$$

where step (a) follows from Hölder inequality and step (b) from the condition imposed in the theorem that $\|\boldsymbol{\varepsilon}\|_1\delta > 3\|\boldsymbol{X}\boldsymbol{\varepsilon}\|_\infty$. Plugging these results back into (S.14) and rearranging

$$\frac{1}{n}\|\boldsymbol{X}\widehat{\boldsymbol{\Delta}}\|_2^2 \leq \frac{\delta}{3n}\|\boldsymbol{\varepsilon}\|_1\|\widehat{\boldsymbol{\Delta}}\|_1 + \frac{\delta}{n}(\|\boldsymbol{\varepsilon}\|_1 - 2\|\boldsymbol{X}\widehat{\boldsymbol{\Delta}} - \boldsymbol{\varepsilon}\|_1)\|\widehat{\boldsymbol{\beta}}\|_1 +$$

$$\frac{\delta}{n}\|\boldsymbol{X}\widehat{\boldsymbol{\Delta}} - \boldsymbol{\varepsilon}\|_1\|\boldsymbol{\beta}^*\|_1 + \delta^2\|\widehat{\boldsymbol{\beta}}\|_1(\|\boldsymbol{\beta}^*\|_1 - \|\widehat{\boldsymbol{\beta}}\|_1).$$

On the one hand:

$$\|\boldsymbol{\varepsilon}\|_1 - 2\|\boldsymbol{X}\widehat{\boldsymbol{\Delta}} - \boldsymbol{\varepsilon}\|_1 \overset{(a)}{\leq} \|\boldsymbol{\varepsilon}\|_1 - \tfrac{3}{2}\|\boldsymbol{X}\widehat{\boldsymbol{\Delta}} - \boldsymbol{\varepsilon}\|_1 \overset{(b)}{\leq} \tfrac{3}{2}\|\boldsymbol{X}\widehat{\boldsymbol{\Delta}}\|_1 - \tfrac{1}{2}\|\boldsymbol{\varepsilon}\|_1,$$

where, in step (a) we use the trivial fact that $-\frac{3}{2} > -2$. The number $\frac{3}{2}$ is arbitrary and other values $-2 < \alpha < -1$ should also work. In step (b), we use the triangular inequality:

$\|X\widehat{\Delta} - \varepsilon\|_1 \geq \|\varepsilon\|_1 - \|X\widehat{\Delta}\|_1$. On the other hand, also using the triangular inequality, we have that: $\|X\widehat{\Delta} - \varepsilon\|_1 \leq \|X\widehat{\Delta}\|_1 + \|\varepsilon\|_1$. Hence:

$$\frac{1}{n}\|X\widehat{\Delta}\|_2^2 \leq \frac{\delta}{3n}\|\varepsilon\|_1\|\widehat{\Delta}\|_1 + \frac{\delta}{n}\left(\tfrac{3}{2}\|X\widehat{\Delta}\|_1 - \tfrac{1}{2}\|\varepsilon\|_1\right)\|\widehat{\beta}\|_1 +$$
$$\frac{\delta}{n}(\|X\widehat{\Delta}\|_1 + \|\varepsilon\|_1)\|\beta^*\|_1 + \delta^2\|\widehat{\beta}\|_1(\|\beta^*\|_1 - \|\widehat{\beta}\|_1).$$

Rearranging the right-hand-side of the above inequality:

$$\frac{1}{n}\|X\widehat{\Delta}\|_2^2 \leq \frac{\delta}{3n}\|\varepsilon\|_1\left(\|\widehat{\Delta}\|_1 + 3\|\beta^*\| - \frac{3}{2}\|\widehat{\beta}\|_1\right) +$$
$$\frac{\delta}{n}\|X\widehat{\Delta}\|_1\left(\|\beta^*\|_1 + \frac{3}{2}\|\widehat{\beta}\|_1\right) + \delta^2\|\widehat{\beta}\|_1(\|\beta^*\|_1 - \|\widehat{\beta}\|_1).$$

Finally, using that the norm inequality : $\|X\widehat{\Delta}\|_1 \leq \sqrt{n}\|X\widehat{\Delta}\|_2$, we obtain

$$\frac{1}{n}\|X\widehat{\Delta}\|_2^2 \leq \frac{\delta}{3n}\|\varepsilon\|_1\left(\|\widehat{\Delta}\|_1 + 3\|\beta^*\| - \frac{3}{2}\|\widehat{\beta}\|_1\right) +$$
$$\frac{\delta}{\sqrt{n}}\|X\widehat{\Delta}\|_2\left(\|\beta^*\|_1 + \frac{3}{2}\|\widehat{\beta}\|_1\right) + \delta^2\|\widehat{\beta}\|_1(\|\beta^*\|_1 - \|\widehat{\beta}\|_1).$$

Notice that the above inequality is a second-order inequality of the type $y^2 \leq by + c$, where $y = \frac{1}{\sqrt{n}}\|X\widehat{\Delta}\|_2$ and $b$ and $c$ can be read by inspection. Since $y \geq 0$ we must have $b^2 + 4c \geq 0$, hence:

$$0 \leq \frac{4}{3}\frac{\delta}{n}\|\varepsilon\|_1\left(\|\widehat{\Delta}\|_1 + 3\|\beta^*\| - \frac{3}{2}\|\widehat{\beta}\|_1\right) + 4\delta^2\|\widehat{\beta}\|_1\left(\|\beta^*\|_1 - \|\widehat{\beta}\|_1\right) + \delta^2\left(\|\beta^*\|_1 + \frac{3}{2}\|\widehat{\beta}\|_1\right)^2$$
$$\leq \frac{4}{3}\frac{\delta}{n}\|\varepsilon\|_1\left(\|\widehat{\Delta}\|_1 + 3\|\beta^*\| - \frac{3}{2}\|\widehat{\beta}\|_1\right) + \delta^2\left(\|\beta^*\|_1^2 + 7\|\widehat{\beta}\|_1\|\beta^*\|_1 - \frac{7}{4}\|\widehat{\beta}\|_1^2\right).$$

Factoring the leftmost term we obtain:

$$0 \leq \frac{4}{3}\frac{\delta}{n}\|\varepsilon\|_1\left(\|\widehat{\Delta}\|_1 + 3\|\beta^*\| - \frac{3}{2}\|\widehat{\beta}\|_1\right) + \delta^2\left(\|\beta^*\|_1 + c_1\|\widehat{\beta}\|_1\right)\left(\|\beta^*\|_1 - c_2\|\widehat{\beta}\|_1\right), \quad \text{(S.15)}$$

where $c_1 = \sqrt{14} + \frac{7}{2} \leq 7.5$ and $c_2 = \sqrt{14} - \frac{7}{2} \geq 0.2$. Using the triangular inequality $\|\widehat{\Delta}\|_1 \leq \|\beta^*\|_1 + \|\widehat{\beta}\|_1$, and the above inequality can be written as

$$0 \leq \frac{4}{3}\frac{\delta}{n}\|\varepsilon\|_1\left(4\|\beta^*\| - \frac{1}{2}\|\widehat{\beta}\|_1\right) + \delta^2\left(\|\beta^*\|_1 + 7.5\|\widehat{\beta}\|_1\right)\left(\|\beta^*\|_1 - 0.2\|\widehat{\beta}\|_1\right).$$

Hence:

$$\|\widehat{\beta}\|_1 \leq 8\|\beta^*\|_1, \quad \text{(S.16)}$$

otherwise, the right-hand side of the inequality would be negative. Now we use the following proposition to obtain a bound on $y^2 = \frac{1}{n}\|X\widehat{\Delta}\|_2^2$.

**Proposition 9.** Let $y, b, c \in \mathbb{R}$ and $b^2 + 4c \geq 0$, if $y^2 \leq by + c$ then $y^2 \leq b^2 + 2c$.

Which yields:

$$\frac{1}{n}\|X\widehat{\Delta}\|_2^2 \leq \frac{2}{3}\frac{\delta}{n}\|\varepsilon\|_1\left(\|\widehat{\Delta}\|_1 + 3\|\beta^*\| - \frac{3}{2}\|\widehat{\beta}\|_1\right) +$$
$$2\delta^2\|\widehat{\beta}\|_1\left(\|\beta^*\|_1 - \|\widehat{\beta}\|_1\right) + \delta^2\left(\|\beta^*\|_1 + \frac{3}{2}\|\widehat{\beta}\|_1\right)^2.$$

Rearranging it we obtain:

$$\frac{1}{n}\|\boldsymbol{X}\widehat{\boldsymbol{\Delta}}\|_2^2 \leq \frac{2}{3}\frac{\delta}{n}\|\boldsymbol{\varepsilon}\|_1\left(\|\widehat{\boldsymbol{\Delta}}\|_1 + 3\|\boldsymbol{\beta}^*\| - \frac{3}{2}\|\widehat{\boldsymbol{\beta}}\|_1\right) + \delta^2\left(\|\boldsymbol{\beta}^*\|_1^2 + 7\|\widehat{\boldsymbol{\beta}}\|_1\|\boldsymbol{\beta}^*\|_1 + \frac{1}{4}\|\widehat{\boldsymbol{\beta}}\|_1^2\right)$$

$$\leq \frac{2}{3}\frac{\delta}{n}\|\boldsymbol{\varepsilon}\|_1\left(\|\widehat{\boldsymbol{\Delta}}\|_1 + 3\|\boldsymbol{\beta}^*\| - \frac{3}{2}\|\widehat{\boldsymbol{\beta}}\|_1\right) + \delta^2\left(\|\boldsymbol{\beta}^*\|_1 + c_1\|\widehat{\boldsymbol{\beta}}\|_1\right)\left(\|\boldsymbol{\beta}^*\|_1 + c_2\|\widehat{\boldsymbol{\beta}}\|_1\right),$$

where $c_1 = \frac{7}{2} + \sqrt{12} \leq 7$ and $c_2 = \frac{7}{2} - \sqrt{12} \geq 0.04$. Now since $\|\widehat{\boldsymbol{\beta}}\|_1 \leq 8\|\boldsymbol{\beta}^*\|_1$ and $\|\widehat{\boldsymbol{\Delta}}\|_1 \leq \|\boldsymbol{\beta}^*\|_1 + \|\widehat{\boldsymbol{\beta}}\|_1 \leq 9\|\boldsymbol{\beta}^*\|_1$,

$$\frac{1}{n}\|\boldsymbol{X}\widehat{\boldsymbol{\Delta}}\|_2^2 \leq \frac{2}{3}\frac{\delta}{n}\|\boldsymbol{\varepsilon}\|_1\left(9\|\boldsymbol{\beta}^*\|_1 + 3\|\boldsymbol{\beta}^*\|\right) + \delta^2\left(\|\boldsymbol{\beta}^*\|_1 + (7\cdot 8)\|\boldsymbol{\beta}^*\|_1\right)\left(\|\boldsymbol{\beta}^*\|_1 + (0.04\cdot 8)\|\boldsymbol{\beta}^*\|_1\right)$$

$$\leq \frac{8}{n}\delta\|\boldsymbol{\beta}^*\|_1\|\boldsymbol{\varepsilon}\|_1 + 76\delta^2\|\boldsymbol{\beta}^*\|_1^2$$

$$\leq 8\delta\|\boldsymbol{\beta}^*\|_1\left(\frac{1}{n}\|\boldsymbol{\varepsilon}\|_1 + 10\delta\|\boldsymbol{\beta}^*\|_1\right).$$

$\square$

*Proof of Proposition 9.* For completeness, we provide proof for the proposition. The inequality can be rewritten as: $-y^2 + by + c \geq 0$ for $b, c \geq 0$. For this type of inequality, it follows that:

$$y \leq \frac{b + \sqrt{b^2 + 4c}}{2}.$$

Therefore:

$$y^2 \leq \left(\frac{b + \sqrt{b^2 + 4c}}{2}\right)^2 \overset{(a)}{\leq} \frac{b^2 + (b^2 + 4c)}{2} = b^2 + 2c,$$

where (a) uses that: $(r + s)^2 \leq 2(r^2 + s^2)$ for any $r, s \geq 0$. $\square$

### C.3 High-dimensional analysis

We follow the same development as [31, example 7.14].

> **Assumption.** Assume $\boldsymbol{\varepsilon}$ has i.i.d. $\mathcal{N}(0, \sigma^2)$ entries and that the matrix $\boldsymbol{X}$ is fixed with maximum entry of $\boldsymbol{X}$ equals to $M$.

**Satisfying the conditions of Theorem 3.** Under our assumptions

$$\max_{j=1,\cdots,m}\frac{\|\boldsymbol{x}_j\|_2}{\sqrt{n}} \overset{(a)}{\leq} \max_{j=1,\cdots,m}\|\boldsymbol{x}_j\|_\infty \leq M.$$

where step (a) follows from the inequality between the norms. Now, $\|\frac{\boldsymbol{X}^\top\boldsymbol{\varepsilon}}{n}\|_\infty$ is a the maximum over $p$ zero-mean Gaussian variable with variance at most $\frac{M^2\sigma^2}{n}$. From standard Gaussian tail bounds:

$$\left\|\frac{\boldsymbol{X}^\top\boldsymbol{\varepsilon}}{n}\right\|_\infty \leq M\sigma\sqrt{\frac{2\log(p/\gamma)}{n}} \tag{S.17}$$

with probability greater than $1 - 2\gamma$. Hence, if we set $\lambda = KM\sigma\sqrt{\frac{\log p}{n}}$ for an appropriate constant $K$ we will have with high probability that $\|\frac{\boldsymbol{X}^\top\boldsymbol{\varepsilon}}{n}\|_\infty \leq \lambda$, satisfying the the condition on Theorem 3.

**Satisfying the conditions of Theorem 2.** Now, we will analyze the condition for which the assumption of Theorem 2 is satisfied. That is, what values of $\delta$ yield with high probability $\|\boldsymbol{X}^\top\boldsymbol{\varepsilon}\|_\infty \leq \delta\|\boldsymbol{\varepsilon}\|_1$. We have that

$$\mathbb{E}\left[\frac{1}{n}\|\boldsymbol{\varepsilon}\|_1\right] = \frac{1}{n}\sum_{i=1}^n \mathbb{E}\left[|\varepsilon_i|\right] \overset{(a)}{=} \sqrt{\frac{2}{\pi}}\sigma,$$

where step (a) relied on the fact that $|\varepsilon_i|$ is a rectified Gaussian variable. Moreover, since the rectified Gaussian is a sub-Gaussian variable with proxy-variance $2\sigma^2$, from a Hoeffding-type of bound

$$\frac{1}{n}\|\varepsilon\|_1 \geq \Big(\sqrt{\frac{2}{\pi}} - 2\sqrt{\frac{\log(1/\gamma)}{n}}\Big)\sigma$$

with probability greater than $1 - 2\gamma$. We combine this result with the (S.17) to obtain that we can set: $\delta = KM\sqrt{\frac{\log p}{n}}$, for an appropriate constant $K$ and we will (with high-probability) satisfy the condition $\|\boldsymbol{X}^\top \varepsilon\|_\infty \leq \delta\|\varepsilon\|_1$.

# D    Numerical experiments

Here we provide some additional descriptions of the numerical experiments.

1. **Isotropic Gaussian features** As mentioned in the main text, we consider Gaussian noise and co-variates: $\epsilon_i \sim \mathcal{N}(0, \sigma^2)$ and $\boldsymbol{x}_i \sim \mathcal{N}(0, r^2 \boldsymbol{I}_p)$ and the output is computed as a linear combination of the features contaminated with additive noise: $y_i = \boldsymbol{x}_i^\top \boldsymbol{\beta} + \epsilon_i$. In the experiments, unless stated otherwise, we use the parameters $\sigma = 1$ and $r = 1$.

2. **Latent-space features model** The "latent space" feature model is described in Hastie *et al.* [26, Section 5.4]. The features $\boldsymbol{x}$ are noisy observations of a lower-dimensional subspace of dimension $d$. A vector in this *latent space* is represented by $\boldsymbol{z} \in \mathbb{R}^d$. This vector is indirectly observed via the features $\boldsymbol{x} \in \mathbb{R}^p$ according to

$$\boldsymbol{x} = \boldsymbol{W}\boldsymbol{z} + \boldsymbol{u},$$

where $\boldsymbol{W}$ is an $p \times d$ matrix, for $p \geq d$. We assume that the responses are described by a linear model in this latent space

$$y = \boldsymbol{\theta}^\top \boldsymbol{z} + \xi,$$

where $\xi \in \mathbb{R}$ and $\boldsymbol{u} \in \mathbb{R}^p$ are mutually independent noise variables. Moreover, $\xi \sim \mathcal{N}(0, \sigma_\xi^2)$ and $\boldsymbol{u} \sim \mathcal{N}(0, \boldsymbol{I}_p)$. We consider the features in the latent space to be isotropic and normal $\boldsymbol{z} \sim \mathcal{N}(0, \boldsymbol{I}_d)$ and choose $\boldsymbol{W}$ such that its columns are orthogonal, $\boldsymbol{W}^\top \boldsymbol{W} = \frac{p}{d}\boldsymbol{I}_d$, where the factor $\frac{p}{d}$ is introduced to guarantee that the signal-to-noise ratio of the feature vector $\boldsymbol{x}$ (i.e. $\frac{\|\boldsymbol{W}\boldsymbol{z}\|_2^2}{\|\boldsymbol{u}\|_2^2}$) is kept constant. In the experiments, unless stated otherwise, we use the parameters $\sigma_\xi = 1$ and the latent dimension fixed $d = 1$.

3. **Random Fourier features model** [43] The features are obtained by the nonlinear transformation $\boldsymbol{z} \mapsto \boldsymbol{x}$:

$$\boldsymbol{x} = \sqrt{\frac{2}{p}}\cos(\boldsymbol{W}\boldsymbol{z} + \boldsymbol{b}),$$

where each entry of $\boldsymbol{W}$ is independently sampled from a normal distribution $\mathcal{N}(0, \sigma_w)$ and each entry from $\boldsymbol{b}$ is sampled from a uniform distribution $\mathcal{U}[0, 2\pi]$ the pair. We apply the random Fourier feature map to inputs of the Diabetes dataset [18]. The outputs are kept unaltered. In the experiments, unless stated otherwise, we use the parameter $\sigma_w = 0.01$.

4. **Random projections model** [40] We consider Gaussian noise and covariates: $\epsilon \sim \mathcal{N}(0, \sigma^2)$ and $\boldsymbol{x} \sim \mathcal{N}(0, \boldsymbol{I}_d)$ and the output is computed as a linear combination of the features contaminated with additive noise: $y = \boldsymbol{x}^\top \boldsymbol{S}^\top \boldsymbol{\beta} + \epsilon$, where $\boldsymbol{S}$ is a random projection matrix of varying dimension. The entries of $\boldsymbol{S}$ are randomly sampled from Rademacher distribution as in the experiments in [40]. In the experiments, unless stated otherwise, we use the $\sigma = 1$.

5. **Phenotype prediction from genotype.** We consider Diverse MAGIC wheat dataset [44] from the National Institute for Applied Botany. The dataset contains the whole genome sequence data and multiple phenotypes for a population of 504 wheat lines. We use a subset of the genotype to predict one of the continuous phenotypes. We have integer input with values indicating whether each one of the 1.1 million nucleotides differs or not from the reference value. Closely located nucleotides tend to be correlated and we consider $\boldsymbol{z}$ a pruned version provided by [44]. To generate the features we subsample $p$ from the sequence $\boldsymbol{z}$, such that the input to the model is $\boldsymbol{x} = \boldsymbol{W}\boldsymbol{z}$, where $\boldsymbol{W}$ is a matrix containing ones or zeros, such that each row of $\boldsymbol{W} \in \mathbb{R}^{p \times d}$ have $p$ nonzero entries, i.e., $\boldsymbol{W}\boldsymbol{1} = p\boldsymbol{1}$.

Examples 1, 2, 4 are synthetic datasets. Examples 3 and 5 are real datasets combined with a feature map strategy. We point out that example 4 requires a (slightly) different mathematical formulation, which we cover in Appendix A.4. The results for the random projection model are presented separately in Figure S.11. The other figures refer to examples 1, 2, 3, 5.

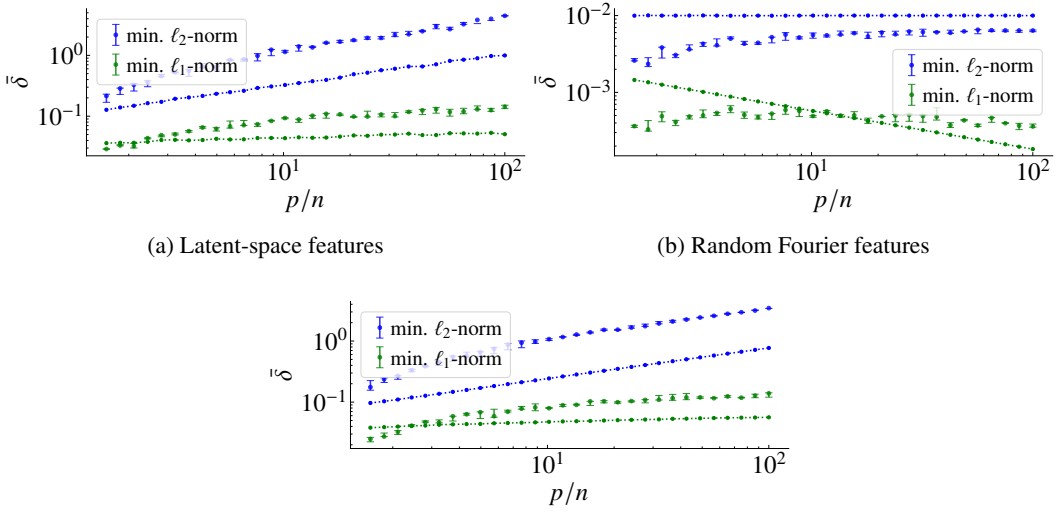

(a) Latent-space features

(b) Random Fourier features

(c) Phenotype prediction from genotype

Figure S.5: **Threshold $\bar{\bar{\delta}}$ vs. number of features.**

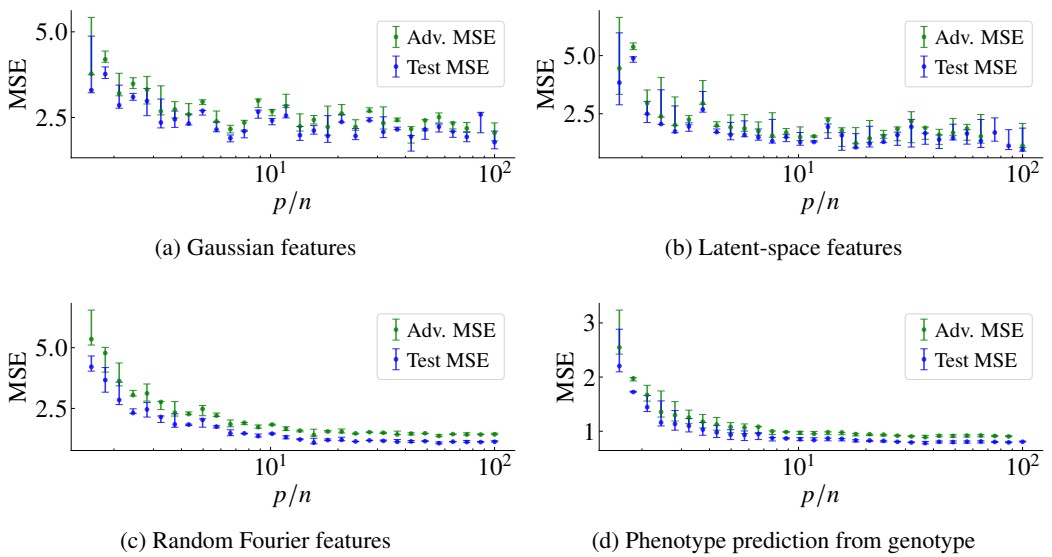

(a) Gaussian features

(b) Latent-space features

(c) Random Fourier features

(d) Phenotype prediction from genotype

Figure S.6: (**Minimum $\ell_2$-norm interpolator**) **MSE on test set *vs.* number of features.** We show both the MSE in the absence of an adversary (Test MSE), and in the presence of an $\ell_2$-adversarial attack (Adv. MSE). The adversarial radius of the evaluation is $\delta_{\text{test}} = 0.01\mathbb{E}\left[\|x\|_2\right]$

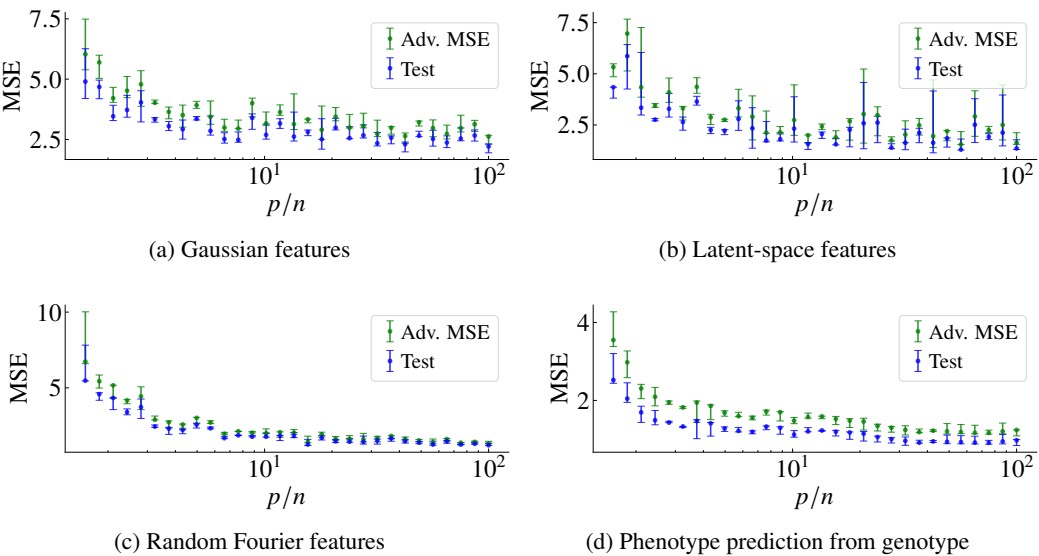

(a) Gaussian features

(b) Latent-space features

(c) Random Fourier features

(d) Phenotype prediction from genotype

Figure S.7: (**Minimum $\ell_1$-norm interpolator**) **MSE on test set *vs.* number of features.** We show both the MSE in the absence of an adversary (Test MSE), and in the presence of an $\ell_\infty$-adversarial attack (Adv. MSE). The adversarial radius of the evaluation is $\delta_{\text{test}} = 0.01\mathbb{E}\left[\|x\|_1\right]$

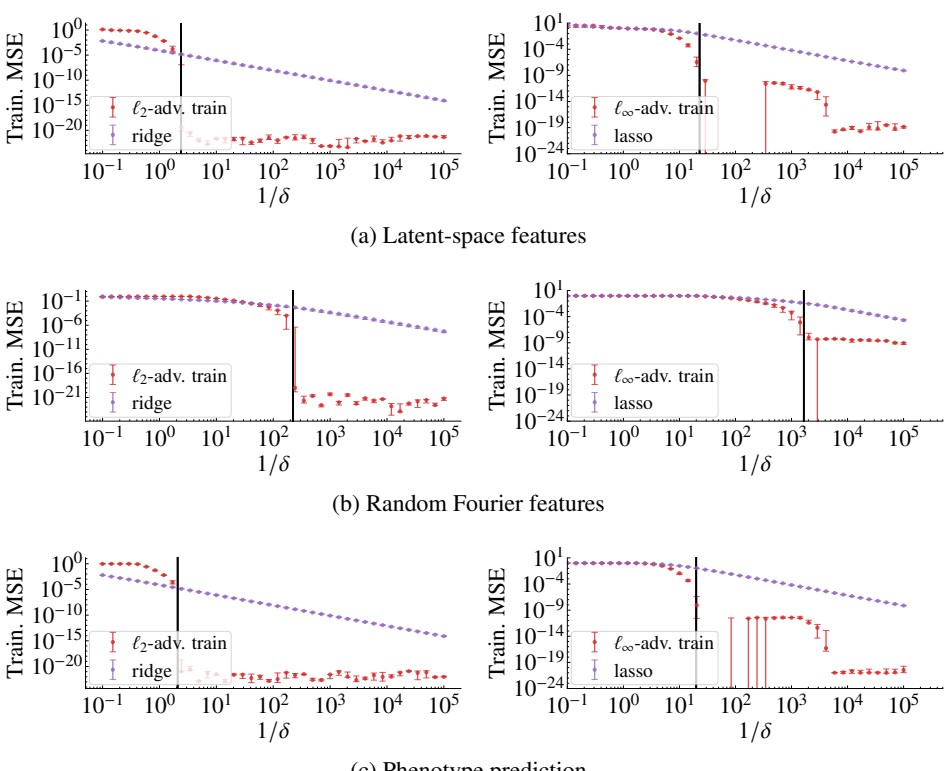

(a) Latent-space features

(b) Random Fourier features

(c) Phenotype prediction

Figure. S.8: **Training MSE *vs* regularization parameter**. *Left:* for ridge and $\ell_2$-adversarial training. *Right:* for Lasso and $\ell_\infty$-adversarial training The error bars give the median and the 0.25 and 0.75 quantiles obtained from numerical experiment (5 realizations).

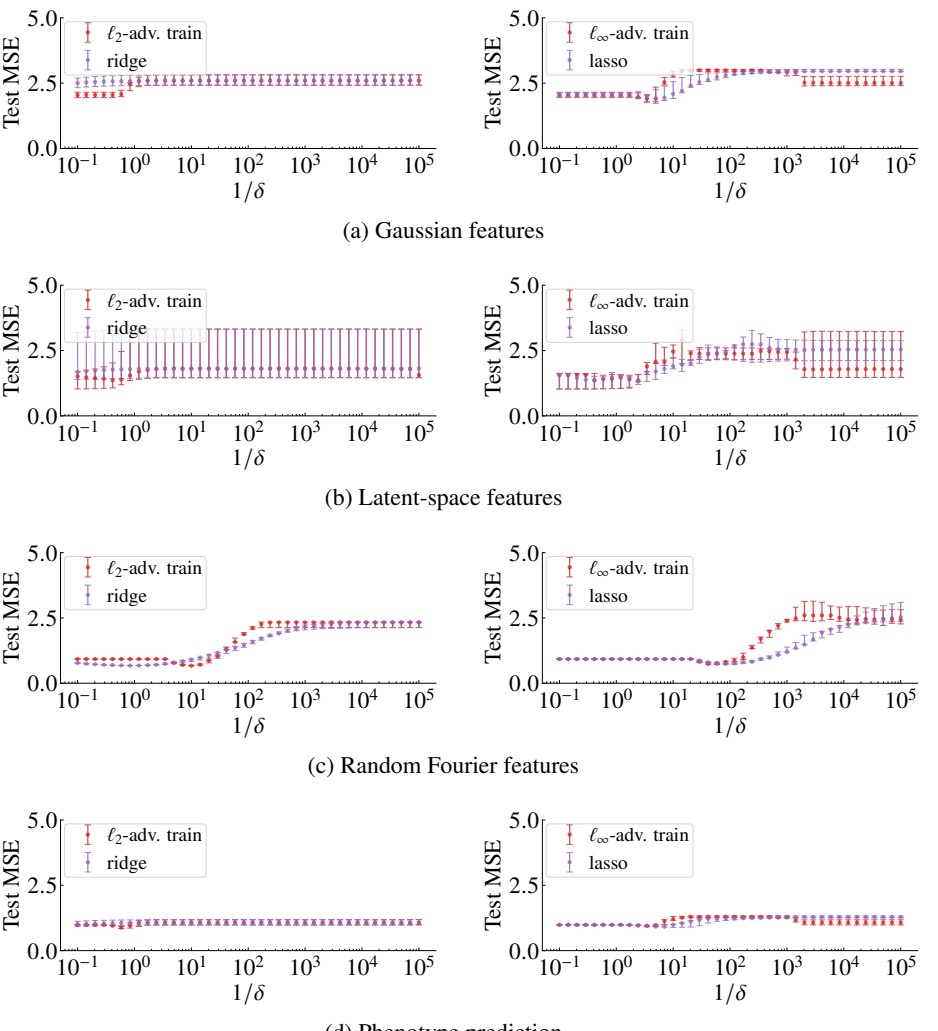

(a) Gaussian features

(b) Latent-space features

(c) Random Fourier features

(d) Phenotype prediction

Figure S.9: **Test MSE *vs* regularization parameter**. *Left:* for ridge and $\ell_2$-adversarial training. *Right:* for Lasso and $\ell_\infty$-adversarial training. The error bars give the median and the 0.25 and 0.75 quantiles obtained from numerical experiment (5 realizations).

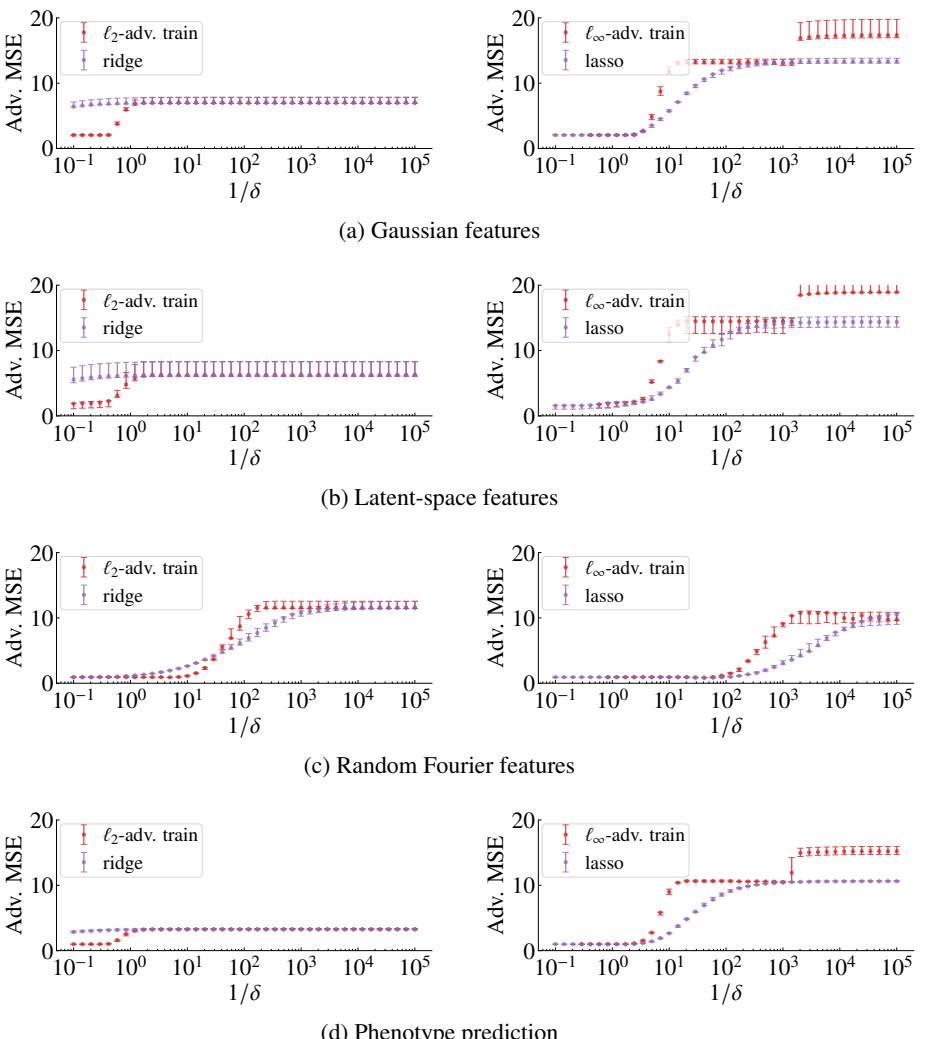

Figure S.10: **Adversarial Test MSE *vs* regularization parameter**. *Left:* for ridge and $\ell_2$-adversarial training. *Right:* for Lasso and $\ell_\infty$-adversarial training. The error bars give the median and the 0.25 and 0.75 quantiles obtained from numerical experiment (5 realizations).

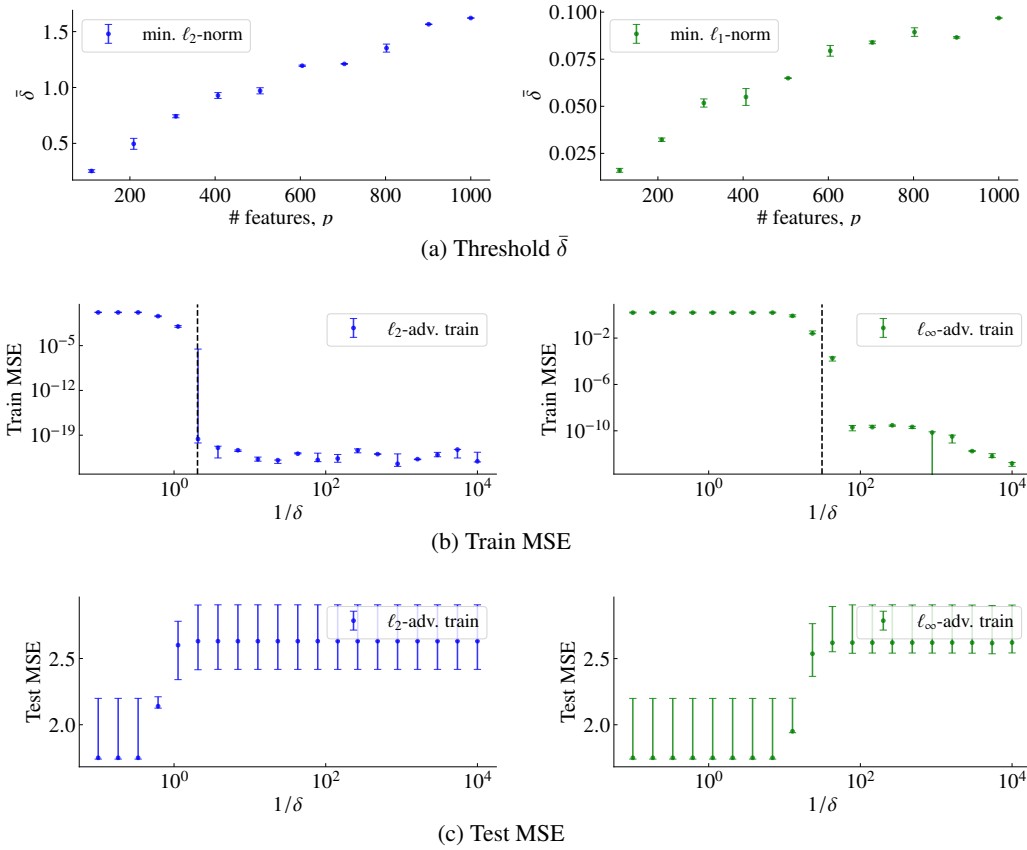

(a) Threshold $\bar{\delta}$

(b) Train MSE

(c) Test MSE

Figure S.11: **Random projections** *Left:* the results for $\ell_\infty$-adversarial training. *Right:* the results for $\ell_2$-adversarial attacks. In (a) we show the threshold as a function of the number of features. Unlike Figures 2 and S.5, we do not give a reference, that is because the input $x$ is fixed, so it does make sense to consider $\delta$ in absolute terms. (b) the train MSE as a function of $1/\delta$ for the number of features fixed $p = 200$. (c) the test MSE as a function of $1/\delta$. We consider an input of dimension $d = 1000$.

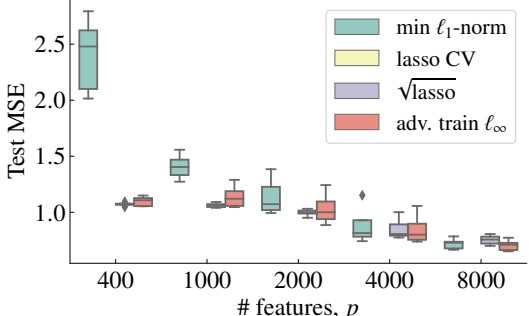

Figure S.12: **Test MSE (MSE) under fixed adversarial radius** in MAGIC dataset. We study a choice of adversarial radius inspired by Theorem 2. For $\ell_\infty$-adversarial training, we use $\delta = 0.5\|\boldsymbol{X}\boldsymbol{\xi}\|_\infty/\|\boldsymbol{\xi}\|_1$ for $\boldsymbol{\xi}$ a vector with zero-mean normal entries. We use a random $\boldsymbol{\xi}$, since we do not know the true additive noise. Even with this approximation, $\ell_\infty$-adversarial training performs comparably with Lasso with the regularization parameter set using 5-fold cross-validation doing a full search among the hyperparameter space. We use a similar method for setting the square-root Lasso parameter, setting $\lambda = 0.1\|\boldsymbol{X}\boldsymbol{\xi}\|_\infty/\|\boldsymbol{\xi}\|_2$. The value 0.5 and 0.1 were set empirically, after finding out that the value 3 used in the theorem was too conservative.

# E Results for general loss functions

## E.1 Proof of Theorem 4

In the proof, we will use the Fenchel conjugate of $L$, where $L^* : \mathbb{R} \to \mathbb{R}$ is defined as:

$$L^*(u) := \sup_z \{uz - L(z)\}$$

*Proof.* Using Fenchel–Moreau theorem the assumption on the loss imply that $L(z) = \sup_u \{uz - L^*(u)\}$ for all $z$. Let $z = \boldsymbol{x}^\top \boldsymbol{\beta}$,

$$
\begin{aligned}
\max_{\|\boldsymbol{\Delta x}\| \leq \delta} \mathcal{L}((\boldsymbol{x} + \boldsymbol{\Delta x})^\top \boldsymbol{\beta}) &= \max_{\|\boldsymbol{\Delta x}\| \leq \delta} \sup_u (u\boldsymbol{x}^\top \boldsymbol{\beta} + u\boldsymbol{\Delta x}^\top \boldsymbol{\beta} - L^*(u)) \\
&= \sup_u \left( u\boldsymbol{x}^\top \boldsymbol{\beta} + \max_{\|\boldsymbol{\Delta x}\| \leq \delta} (u\boldsymbol{\Delta x}^\top \boldsymbol{\beta}) - L^*(u) \right) \\
&= \sup_u \left( u\boldsymbol{x}^\top \boldsymbol{\beta} + \delta \|\boldsymbol{\beta}\|_* |u| - L^*(u) \right) \\
&= \max_{s \in \{-1,1\}} \sup_u \left( u(\boldsymbol{x}^\top \boldsymbol{\beta} + s\delta \|\boldsymbol{\beta}\|_*) - L^*(u) \right)
\end{aligned}
$$

Applying Fenchel–Moreau theorem again we obtain the desired result. $\qquad\square$

## E.2 Extension to linear maps

We consider here a matrix $\boldsymbol{S} \in \mathbb{R}^{p \times d}$ maps from the input space $\mathbb{R}^d$ to a feature space $\mathbb{R}^p$ (The setting described in Appendix A.4). And that the parameter vector is estimated in this features space. In this case, the following extension holds:

> **Theorem 7.** Let $\mathcal{L} : \mathbb{R} \to \mathbb{R}$ be convex and lower-semicontinuous, than for every $\delta$
>
> $$\max_{\|\boldsymbol{\Delta x}\| \leq \delta} \mathcal{L}((\boldsymbol{x} + \boldsymbol{\Delta x})^\top \boldsymbol{S}^\top \boldsymbol{\beta}) = \max_{s \in \{-1,1\}} \mathcal{L}\left(\boldsymbol{x}^\top \boldsymbol{S}^\top \boldsymbol{\beta} + \delta s \|\boldsymbol{S}^\top \boldsymbol{\beta}\|_*\right). \tag{S.18}$$

the proof follows the same steps and is omitted here.

## E.3 Application to dimensionality reduction

In this example, we discuss how to formulate an adversarial dimensionality reduction algorithm from principal component analysis (PCA) loss function. PCA finds a projection matrix $\boldsymbol{P}$ that transforms a given input $\boldsymbol{x} \in \mathbb{R}^m$ into a lower-dimensional representation $\boldsymbol{z} = \boldsymbol{P}\boldsymbol{x} \in \mathbb{R}^d$, with $d < m$. Conversely, given a lower-dimensional representation, the original input space can be reconstructed with the inverse transformation $\hat{\boldsymbol{x}} = \boldsymbol{P}^\top \boldsymbol{z}$. Principal components analysis can be formulated as the minimization of

$$\frac{1}{n} \sum_{i=1}^n \|\boldsymbol{x}_i + \boldsymbol{P}_d \boldsymbol{P}_d^\top \boldsymbol{x}_i\|_2^2. \tag{S.19}$$

for $\boldsymbol{P} \in \mathbb{R}^{m \times d}$ a matrix with orthonormal columns $\boldsymbol{P}_d = [\boldsymbol{p}_1, \cdots, \boldsymbol{p}_d]$. Alternatively, it can be viewed as sequentially minimizing:

$$\frac{1}{n} \sum_{i=1}^n \|\tilde{\boldsymbol{x}}_i + \boldsymbol{p}_d \boldsymbol{p}_d^\top \tilde{\boldsymbol{x}}_i\|_2^2 \quad \text{subject to} \quad \boldsymbol{P}_{(d-1)} \boldsymbol{p}_d = 0 \text{ and } \|\boldsymbol{p}_d\|_2 = 1 \tag{S.20}$$

where $\tilde{\boldsymbol{x}}_i = \boldsymbol{x}_i - \boldsymbol{P}_{(d-1)} \boldsymbol{P}_{(d-1)}^\top \boldsymbol{x}_i$. Moreover, minimizing (S.20) is equivalent to minimize $\frac{1}{n} \sum_{i=1}^n \mathcal{L}(\tilde{\boldsymbol{x}}_i^\top \boldsymbol{p}_d)$ for $\mathcal{L}(\tilde{\boldsymbol{x}}^\top \boldsymbol{p}_d) = -(\tilde{\boldsymbol{x}}^\top \boldsymbol{p}_d)^2$. With the formulation of PCA we just described, the adversarial extension follows naturally. One can obtain that for this loss function $s_* = -\text{sign}(\tilde{\boldsymbol{x}}^\top \boldsymbol{p}_d)$ and that

$$\max_{\|\boldsymbol{\Delta x}\| \leq \delta} \|(1 + \boldsymbol{p}_d \boldsymbol{p}_d^\top)(\tilde{\boldsymbol{x}} + \boldsymbol{\Delta x})\|_2^2 = -\left(|\tilde{\boldsymbol{x}}^\top \boldsymbol{p}_d| - \delta \|\boldsymbol{p}_d\|_*\right)^2.$$

For the special case of $\ell_2$-adversarial disturbance, $\|\boldsymbol{p}_d\|_2 = 1$ hence:

$$\max_{\|\boldsymbol{\Delta x}\|_2 \leq \delta} \|(1 + \boldsymbol{p_d p_d}^\top)(\tilde{\boldsymbol{x}} + \boldsymbol{\Delta x})\|_2^2 = -\left(|\tilde{\boldsymbol{x}}^\top \boldsymbol{p_d}| - \delta\right)^2.$$