# OpenReview forum: "Regularization properties of adversarially-trained linear regression"
_NeurIPS.cc/2023/Conference — NeurIPS 2023 spotlight_

### Official Review · Reviewer_uRwm · 2023-06-13

**Soundness:** 4 excellent
**Presentation:** 4 excellent
**Contribution:** 3 good
**Rating:** 8
**Confidence:** 4

**Summary:**

This paper provides an in-depth analysis of adversarial training with linear models and its relationship to regularized regression methods under the overparametrized regime.
Depending on the value of the perturbation radius, it is revealed that there exist three modes.
When the radius is small, solutions to adversarial training behave as minimum-norm interpolators (Theorem 1).
When the radius is medium, the solutions behave as solutions to the parameter shrinkage regression (Proposition 4).
When the radius is large, the zero solution is necessary and sufficient (Proposition 3).
In addition to these theoretical results, the authors observe the mode change experimentally and discuss how adversarial training is advantageous over parameter shrinkage regression.

**Strengths:**

- A modern extension of theory on robust optimization and regularization: The relationship between robust optimization (somewhat encompassing adversarial training in this work) and regularization has been known in the literature, including Xu et al. (2009). This work contributes to studying what happens when it comes to overparametrization and nicely characterizes the relationship between the perturbation radius and the corresponding modes (as I summarized above).
- Demonstration of the benefit of overparametrization: In the numerical simulation of Figure 2, the authors demonstrate that the robustness radius increases as the model becomes more overparametrized, namely, $p/n$ increases. This clearly indicates the benefits of overparametrization (though the analysis hinges on norm matching, as mentioned in Remark 2).
- Clarity: Despite the thorough theory, the paper is written clearly and easy to follow.

Xu et al. (2009). "Robustness and Regularization of Support Vector Machines." (JMLR)

**Weaknesses:**

One of the main weaknesses would be the restriction to linear models, which is crucial for the current analysis yet needed for further understanding adversarial training.

You may refer to Xu et al. (2009) when you show Theorem 4. Indeed, the equation right after l.322 can be regarded as a generalization of Theorem 3 in Xu et al. (2009) because $\\ell(y(\\boldsymbol{x}^\\top\\boldsymbol{\\beta}) - \\delta\\|\\beta\_\*\\|) \\le \\ell(y(\\boldsymbol{x}^\\top\\boldsymbol{\\beta})) + \\delta\\|\\boldsymbol{\\beta}\\|\_\*$ when $\\ell$ is the hinge loss.

Below, I have other minor comments.

- In Figure 1, can you specify what $\\lambda$ and $\\delta$ are used for each line?
- In the proof of Theorem 1, you may need $-$ (negative) sign in front of either $\\epsilon\_i\boldsymbol{x}\_i$ in Eq. (6) or $n\\delta\boldsymbol{\\alpha}$ in l.132. Otherwise, "the subderivative contains zero" (l.132) does not seem to be correct.
- In l.150, the reason of $\\delta\_{\\text{test}} \\propto \mathbb{E}[\\|\\boldsymbol{x}\\|]$ is unclear to me. Can you elaborate on it?
- In Figure 4, can you specify what $n$ and $p$ are used?
- In Eq. (8), do you miss the exponent $2$ for the norm?
- In Eq. (9), it might be better to change the notation $\\epsilon$ for the noise because $\\epsilon$ has already been used in the proof of Theorem 1.
- In the equations after l.319 and l.322, should we need $+ \\Delta x$ on the left-hand sides?
- In the appendix, what is referred to as Theorem 3 seems to be Proposition 3.

**Questions:**

See the weaknesses.

**Limitations:**

Obviously, the analysis is entirely limited to the linear model case. Nonetheless, the analysis provides a fair amount of insights to readers, so I don't think this is a big limitation.

---

> ### Author Rebuttal · Authors · 2023-08-08
>
> We greatly appreciate the reviewer's comments and feedback. We are thankful for the overall positive assessment of our work and also believe the comments will greatly improve the paper.
>
> **About Theorem 4 and its relation to Xu et al (2009).** Thank you for the comment. We updated the manuscript including the reference and its connection with our Theorem 4.
>
> **Other minor comments.** (we answer in order)
> - Good suggestion. There is a one-to-one correspondence between the x-axis in Figure 1 (i.e., $||\widehat \beta||\_1$) and the value of $\lambda$ or $\delta$. We will add the figure displaying the relation in the Supplementary material (and add a pointer to it in the caption of Fig.1).
> - Indeed, it should have been $\varepsilon = - n \delta \widehat{\alpha}$. Thank you for catching it.
> - Thank you for the question.  In the end, the relative size of the adversarial attacks $||\Delta x||$ in relation to $||x||$ is what matters, not its absolute value (i.e., having a very small disturbance $\Delta x$ will have a different meaning when all the inputs are also very small, compared to when they are normalized). In Figure, as we increase $p$ the magnitude of the inputs $||x||$ can also increase depending on how the features are added. By making $\delta\_{\text{test}} \propto E[||x||]$ we make sure this effect is taken into account.
> - We use $n=60$, $p =200$. We will add this information to the figure caption.
> - No, it is correct. Nonetheless, you have a good point: the optimization problem with the exponent 2 for the norm could also work (since it is an equivalent problem). We use without the exponent so it is the same formulation as in reference [30].
> - Good point. We will choose a different letter that does not override the notation.
> - We believe the equations are correct. These two equations are specific cases of eq. (11). And indeed, in Theorem 4 the double application of Fenchel duality allows us to get rid of $\Delta x$ in the left-hand side of (11).
> - Yes, you are correct. Thank you for spotting it.

---

> > ### Comment · Reviewer_uRwm · 2023-08-15
> > **Response**
> >
> > Thank you for the response. The authors' elaboration mostly cleared up my questions.
> > On my minor comment
> >
> > > In the equations after l.319 and l.322, should we need $+\\Delta x$ on the left-hand sides?
> >
> > I just wanted to draw attention to possible typos in the left-hand sides; for example, $\\max_{\\|\\Delta x\\| \\le \\delta}\\ell(|x^\\top \\beta-y|)$ (right after l.319 in the original version) seems to be $\\max_{\\|\\Delta x\\| \\le \\delta}\\ell(|(x+\\Delta x)^\\top \\beta-y|)$, otherwise there is no $\\Delta x$ to be maximized, which does not align with the left-hand side of eq. (11).
> >
> > If I still misunderstand anything, feel free to point out.

---

> > > ### Author Response · Authors · 2023-08-16
> > >
> > > You are absolutely correct. We misunderstood your initial comment... Thank you very much for clarifying. We will fix it

---

### Official Review · Reviewer_Zg7q · 2023-06-16

**Soundness:** 3 good
**Presentation:** 3 good
**Contribution:** 3 good
**Rating:** 6
**Confidence:** 4

**Summary:**

This paper studies the connection between adversarial training and regularization methods in linear regression problem. Simulation studies are provided to justify the correctness of their theoretical observations.

**Strengths:**

The authors conducts a comprehensive study on the relationship between adversarial training and regularization methods in linear regression setup. The writing is clear and easy to understand.

**Weaknesses:**

My major concern towards this paper is the limit of its contribution. While the analysis is comprehensive, it is only restricted to linear models. Considering that the adversarial training is more commonly used in neural networks rather than linear models in reality, the contribution is limited. The authors are encouraged to add more discussions on neural networks.

In addition, the following paper considers the connection between regularization and adversarial robustness:

Jakubovitz, Daniel, and Raja Giryes. "Improving dnn robustness to adversarial attacks using jacobian regularization." Proceedings of the European Conference on Computer Vision (ECCV). 2018.

Please cite this paper and compare it to the submission from intuition aspect.

**Questions:**

Is it possible to extend the analysis to two-layer neural networks?

---

> ### Author Rebuttal · Authors · 2023-08-08
>
>
> We thank the reviewer for useful feedback and comments.
>
> **Why restrict the analysis to linear models?** We understand your concern and believe you raise a very valid point: adversarial training is very often used in the context of neural networks and it is natural to be interested in the behavior of this class of models.
>
> We will try to have a more extensive discussion on the consequences of our results to neural networks. While neural networks are not analyzed, we cover some of the simplified theoretical models commonly used to analyze neural networks. For instance, Random Fourier features models can be analyzed using our theory and can be seen as a simplified neural network model (one-layer neural network where only the last layer weights are adjusted), see Section 7, example 5.  Moreover, we will try to clearly highlight possible uses of our results for analyzing neural networks. For instance, in Appendix A.4., we provide adversarial training after linear projections, and could for instance be used to analyse deep linear networks, as the ones studied in [A1, A2]. Section 8 provides a result for infinite spaces, and one could attempt to use it to analyze infinitely-wide neural networks, as the ones studied in [A3, A4].
>
> We admit that making these connections might help, but still has limited effect in addressing this limitation. Nonetheless, we believe the analysis of linear models is rich enough to justify an interest of its own and does not necessarily reduce the impact or relevance of the paper:
> 1. A thorough analysis of adversarial attacks in linear models is a natural toolbox for any researchers trying to understand nonlinear models.
> 2. We believe the results presented in this paper are novel and provide a hint at many interesting phenomena that can be observed in neural networks. Providing, for instance, insight into the relation between overparameterization and robustness.
> 3. There are a lot of extensions and future work that can be done from this paper. We hope our group and other groups can use it as a starting point for further analysis.
>
> **On the connection with the ECCV paper.** Thank you very much for the reference. We will cite and add a comment establishing the connection. The paper is indeed relevant, it establishes a connection between adversarial robustness and regularization methods in more complex scenarios than studied by us. We establish a direct connection between regularization and adversarial training in linear models. This idea is complemented by your reference, which shows how the interplay between robustness and regularization is still relevant in deep neural networks.
>
> **References**
> - [A1] A. M. Saxe, J. L. McClelland, and S. Ganguli, “Exact solutions to the nonlinear dynamics of learning in deep linear neural networks,” ICLR, 2014.
> - [A2] T. Laurent and J. Brecht, “Deep Linear Networks with Arbitrary Loss: All Local Minima Are Global,” in ICML 2018.
> - [A3] J. Lee et al., “Wide neural networks of any depth evolve as linear models under gradient descent,” J. Stat. Mech., vol. 2020, no. 12, p. 124002, Dec. 2020, doi: 10.1088/1742-5468/abc62b.
> - [A4] A. Jacot, F. Gabriel, and C. Hongler, “Neural Tangent Kernel: Convergence and Generalization in Neural Networks,” NeurIPS, 2018

---

> > ### Comment · Reviewer_Zg7q · 2023-08-12
> >
> > Thanks for your response.

---

### Official Review · Reviewer_6WVQ · 2023-07-03

**Soundness:** 2 fair
**Presentation:** 2 fair
**Contribution:** 2 fair
**Rating:** 6
**Confidence:** 3

**Summary:**

The paper studies adversarial training (AT) for linear regression for which the inner maximization problems has a closed form solution. They then attempt at relating the solutions to solutions of other optimization problems:

- They show that the minimum norm interpolator also minimizes the adversarial loss (iff the adversarial perturbation is sufficiently small)
- They show that adversarial training (under certain conditions) minimizes something closely related to the LASSO and ridge regression objective for $\ell_\infty$ and $\ell_2$ attacks respectively.
- They show that similarly to square-root LASSO, adversarial training does not need knowledge of the variance and they argue that this makes adversarial training a viable alternative.


**Strengths:**

- It seems interesting to attempt connecting AT to sparse solutions
- The initial setup and the statements of the theorems are presented in a clean way
- Existing literature is well-covered


**Weaknesses:**

- My main concern is that the theoretical claims are rather weak:
    - Concerning Thm. 1, l. 122 "minimum-norm interpolators as the outcome of adversarial training" seems a bit of a stretch, since it is not *consistently* the outcome of adversarial training (we might be able to find a minimizer of $R^{adv}$ that is *not* a min norm interpolator). AT would imply minimum-norm interpolator if the minimizer of $R^{adv}$ was unique, but this cannot be the case since LASSO is not unique in general.
    - Prop. 2 is concerning minimum norm interpolator (so not necessarily obtainable with AT!). What makes this statement interesting for adversarial training if we need to obtain the solution through other means?
    - Prop. 4 seems to not directly relate AT to LASSO/ridge regression. Whats is the conclusion of Prop. 4?
    - Thm. 2 exists to show that AT can replace sqrt-root LASSO. You are comparison with Lasso though – doesn't the bound have a bias in comparison with sqrt-root LASSO (eq. 10 of [29]). The main feature of AT seems to be the claim that $\delta^*$ is invariant to rescaling of $\varepsilon$. Can you explicitly make $\delta^*$ in Thm. 2 independent of $\varepsilon$? (currently this is not the case in theorem statement)

Comments:

- Prop. 5 maybe pick a different variable than $p$ (already used for dimensionality)
- l. 144 should have been $\delta$ instead of $\delta_{train}$?
- Maybe write "a solution" in l. 144 instead of "the".
- l. 179: Please describe the dataset in the appendix or provide a more direct pointer to [18].


**Questions:**

- Thm. 1: $\bar \delta$ depends on the $\ell_\infty$-norm regardless of the choice of norm in the adversarial training? This seems potentially loose – could you comment on it?
- Prop. 2: Do you still rely on full row rank in Prop. 2?
- l. 158-159: Isn't the claim in [17] about $\ell_2$ minimum norm while your Prop. 7 is a claim about choice of norm in the adversarial training?
- Figure 3: Could you label the plot to explain the colors? I don't understan how to interpret the plot.
- Figure 4 / l. 179: what is "regularization paths"?
- What assumption breaks in Prop. 4 since it is no longer able to predict similarly after $\delta$ is made sufficiently small (as demonstrated in Fig. 4)?

---

> ### Author Rebuttal · Authors · 2023-08-08
>
> We thank the reviewer for the detailed feedback. We are grateful the reviewer pointed out aspects that were unclear. We answer your comments below and will update the paper accordingly.
>
> - **Thm 1:**  Indeed, we did not establish uniqueness. We will tone down our claims and rewrite so it is not misleading.
> - **Prop 2:** Proposition 2 provides a way of analyzing the adversarial robustness of minimum-norm interpolators in test time.  Theorem 1 allows us to see minimum-norm interpolators as *a* solution to adversarial training, giving a new perspective on its robustness. The results, however,  only refer to training performance. Proposition 2 adds to that picture by giving intuition about how $\bar \delta$ affects the adversarial robustness in *test-time* (which is what we are ultimately interested in).
> - **Prop 4:**  Thank you for the comment, we use this opportunity to clarify it (it also generated a question from Reviewer CYXN). A possible interpretation is to view
> $$\min\_{\beta}\frac{1}{n} \sum\_{i=1}^n (y\_i - x\_i^\top \beta)^2 + \Big(\delta ||\beta||\_1 + \frac{1}{n}||y||\_1\Big)^2,$$
> (in Proposition 4) as the Lagrangian formulation of the following constrained optimization problem
> $$\min\_{\beta} \frac{1}{n} \sum\_{i=1}^n (y\_i - x\_i^\top \beta)^2\text{ subject to }\Big(\delta ||\beta||\_1 + \frac{1}{n}||y||\_1\Big)^2 \le \Delta,$$
> for some $\Delta$. We can then rewrite the constraint as
> $$\delta ||\beta||\_1 \le \sqrt{\Delta} - \frac{1}{n}||y||\_1.$$
> Note that we can always rescale the data so that $\sqrt{\Delta} - \frac{1}{n}||y||\_1 > 0$ and the problem is well-posed. In turn, the Lagrangian formulation of the modified problem is
> $$\min\_{\beta}\frac{1}{n} \sum\_{i=1}^n (y\_i - x\_i^\top \beta) +  \lambda' ||\beta||\_1.$$
> for some $\lambda'$ .  After this rewriting, it becomes clear that Proposition 4 establishes the equivalence between $\ell\_\infty$-adversarial training and lasso (but $\lambda'$ does not have a closed-formula expression). A similar reasoning could be used to connect the result for $\ell\_2$-adversarial training with ridge regression. We will make it more clear in the revised text.  The equivalence is also observed experimentally in Figs 1 and 3.
> - **Thm 2:** Theorem 2 establishes similarities between $\ell\_\infty$-adversarial training and square-root lasso.  Our derivation yield the same convergence rates as would be obtained under the same assumptions for square-root lasso, any additional term that appears in the derivation vanishes faster as we set $\delta \propto R \sqrt{(\log p) / n}$.  We do not believe that the theorem implies that adversarial training replaces square-root lasso: for both methods, we can set the regularization parameter without knowledge of the variance and get the same rate of convergence.
>
>   Also, as mentioned in Remark 5, Eq. (10) in [28] is not directly comparable to ours, we bound the prediction error, while they bound the parameter estimation error under data assumptions that differ from ours.
>
>   We are not aware of any way to make $\delta^*$ independent of $\epsilon$, neither for $\ell\_\infty$- adversarial training nor square-root lasso. The results where $\epsilon$ does directly not appear in [28] follow from using subgaussian upper bounds (that also could be used in our case).
>
> **Comments**
> - Indeed, $p$ was a bad choice since it overrides the notation. We will change it.
> - Using $\delta\_{\text{train}}$ was intentional. The reason to use it was to make the contrast with $\delta_{\text{test}}$ more clear. We will add a sentence to clarify the notation.
> - Good point, without proving uniqueness we cannot use "the". We will rewrite the sentence.
> - We will add a subsection in the appendix with a short description of the dataset and point to it.
>
> **Questions**
> - The choice of the adversarial training norm appears indirectly: it determines which dual problem $\widehat{\alpha}$ solves.  We highlight that the result is tight: the theorem is an "if and only if" statement. We also check it experimentally in Fig 4, there the model interpolates (up to numerical precision) the training data for solutions where $\delta \le \bar \delta$ with the transition happening exactly at $\bar{\delta}$.
> - It is necessary. We will update the text. This was implicit since without full rank there exists no interpolating solution. But, we agree that it is more clear to explicitly say it.
> - Proposition 7 is about the gap between training and testing for minimum-norm interpolators. Theorem 1 allows us to see the minimum-norm solution as *an* optimal solution during adversarial training. Proposition 7 shows that despite being optimal during training this solution can still have very large adversarial test error for mismatched evaluations. What, for instance, allow us to explain Fig 1 in [17].
> - In Fig  3, we are considering $p=10$  coefficients, corresponding to the ten variables in the problem (age, sex, body mass index, average blood pressure, and six blood serum measures). Each color corresponds to the value of one coefficient of the coefficients $\beta_i$ in the regression problem, as we change the value of the regularization. We will include a legend.
> - We will make it more clear in the text. The regularization path is a plot of the coefficient value against varying values of the regularization parameter (for adversarial training the adversarial radius $\delta$, for ridge and Lasso $\lambda$). This type of analysis is used in the study of Lasso and variants, see [17] and [B1, Section 3.4].
> - The hypothesis $||\widehat{\beta}||\ge \min_i \frac{|y_i|}{\|x_i\|}$ no longer holds after the transition point in Fig 4. That is because after the transition point $y_i = \widehat{\beta}^\top x\_i$ for all $i$. Hence, $|y\_i| = |\widehat{\beta}^\top x\_i| \ge ||\widehat{\beta}||_*||x\_i||$, with the strict inequality holding for some $i$.
>
> **References**
> - [B1] J. Friedman, T. Hastie, and R. Tibshirani *"The elements of statistical learning"* 2001.

---

> > ### Comment · Reviewer_6WVQ · 2023-08-17
> >
> > Thank you for the elaborate response which clarifies a lot. I'm mostly happy with the response and have raised my score, even though I still remain convinced that the theoretical statements regarding AT are rather weak. I would just ask you to include the following:
> >
> > - Thm 1 & Prop 2: Since uniqueness is not established in Thm 1, we cannot  use the statement to draw conclusions about AT from characterization of minimum-norm interpolators (Prop 2). Because of that, apart from toning down the claim regarding Thm 1, I would maybe also explicitly clarify that Prop 2 is not characterizing AT (since AT is otherwise the focus of the paper).
> >
> > - Prop 4: The last equation you obtain in your rebuttal after rewriting the proposition contains
> > $\|\beta\|_1$.
> >
> >     I guess this should have been $\|\beta\|_\infty$. Including the form that you provided in your rebuttal and explicitly commenting on $\lambda'$ would make the connection more precise.

---

> > > ### Author Response · Authors · 2023-08-20
> > >
> > > Thank you very much for taking the rebuttal into consideration and raising your score.  Also, thank you for the two suggestions: we will clarify Proposition 2 and add the connection in Prop 4 (with the ideas we wrote in the rebuttal).

---

### Official Review · Reviewer_CYXN · 2023-07-04

**Soundness:** 3 good
**Presentation:** 3 good
**Contribution:** 3 good
**Rating:** 6
**Confidence:** 4

**Summary:**

This paper investigates adversarial training of linear regression. The authors compared the solution of adversarial training and other regularization frameworks (minimum-norm interpolating, ridge regression, Lasso and square-root Lasso), and established close relations between adversarial training and other methods under certain conditions depending on the disturbance radius and over/under-parameterization. The authors also consider extending the result to more general loss function for linear model.

**Strengths:**

1.	The paper provides valuable insights on the relation between adversarial training and other regularization frameworks for linear regression, which contributes to the area of robust learning. The analysis is sound.
2.	The paper provides good background knowledge and details in their work.
3.	The paper is well-organized and easy to follow overall.


**Weaknesses:**

1.	In the abstract, the authors claim that adversarial training can be equivalent to parameter shrinkage methods (like ridge regression and Lasso). However, from Proposition 4, it seems the two frameworks are not equivalent, since the regularization term in the equation of Proposition 4 equals $\delta^2\left\| \beta\right\|^2 + c\delta\left\| \beta\right\|$ for some constant $c$. I am curious about how the quadratic term can affect the solution, or how close the adversarial training solution is from the parameter shrinkage solution.

2.	In the numerical experiment, the authors have not mentioned how the adversarial training is carried out in these datasets. From the code in the supplementary materials, it seems the adversarial samples are generated by the PGD attack. Please consider including more details in the paper. Also, does PGD generate sufficiently strong attacks for linear regression?


**Questions:**

Here are some additional questions/comments:

1.	$\sigma_1$ and $\sigma_n$ in line 126 are undefined.

2.	The authors claim in line 151 and Remark 2 that the model becomes robust as feature dimension $p$ grows, which seems not precise to me. The authors suggest that the threshold $\bar{\delta}$ increases faster, but this only guarantees that the optimal solution of adversarial training and minimum-norm interpolator agree, which does not necessarily mean more robustness. Does the risk $\mathcal{R}^{\text{adv}}$ decrease as feature dimension $p$ grows?

3.	The paper investigates the situation where the sample features $x_i$’s are disturbed in linear regression. In applications, it is also very common that the target $y_i$’s are disturbed.


**Limitations:**

The authors have adequately addressed the limitations.

---

> ### Author Rebuttal · Authors · 2023-08-08
>
> We thank the reviewer for the careful reading of our paper (including even going into the code). We answer the comments below.
>
> 1.  **On the equivalence with parameter shrinking methods.** Thank you for the comment. A possible interpretation is to view
> $$\min\_{\beta}\frac{1}{n} \sum\_{i=1}^n (y\_i - x\_i^\top \beta)^2 + \Big(\delta ||\beta||\_1 + \frac{1}{n}||y||\_1\Big)^2,$$
> (in Proposition 4) as the Lagrangian formulation of the following constrained optimization problem
> $$\min\_{\beta} \frac{1}{n} \sum\_{i=1}^n (y\_i - x\_i^\top \beta)^2\text{ subject to }\Big(\delta ||\beta||\_1 + \frac{1}{n}||y||\_1\Big)^2 \le \Delta,$$
> for some $\Delta$. We can then rewrite the constraint as
> $$\delta ||\beta||\_1 \le \sqrt{\Delta} - \frac{1}{n}||y||\_1.$$
> Note that we can always rescale the data such that  $\sqrt{\Delta} - \frac{1}{n}||y||\_1 > 0$ so that the problem is well-posed. In turn, the Lagrangian formulation of the modified problem is
> $$\min\_{\beta}\frac{1}{n} \sum\_{i=1}^n (y\_i - x\_i^\top \beta) +  \lambda' ||\beta||\_1.$$
> for some $\lambda'$ . After this reformulation, the equivalence between $\ell\_\infty$-adversarial training and lasso becomes explicit, but the value of $\lambda'$ does not have a closed-form expression. A similar reasoning could be used to connect the results of $\ell\_2$-adversarial training with ridge regression. We will clarify this in the text. We also highlight that the equivalence is corroborated experimentally in Figures 1 and 3 (the regularization paths are very similar).
> 2. **On the use of PGD.** We apologize that the presence of PGD implementation caused confusion. The algorithm was only used for double-checking some of the results. In the case of linear regression, we can generate the attacks exactly (we use the function `compute_adv_attack` in the supplementary file `code/adversarial_attack.py`). The function is based on the results of Theorem 4 and gives the optimal solution to the maximization problem. We will add additional comments on the code to make it clearer. We will also add an explanation in the paper to make sure the experimental setup is clear.
>
>
> **Questions:**
> 1. Thank you for catching it. Here, $\sigma\_1\le \cdots \le \sigma\_n$ are the eigenvalues. We will make sure that it is defined in the revised paper.
> 2. Yes, it does decrease. We will add the plot in the supplementary material. The intuition behind it is that each time you add one more feature to your model the previous solution is still a valid one, so the training error would either be the same (if the weight given to the parameter that multiplies this feature is zero) or decrease (if there is a value that produces a better solution).
> 3. Interesting point. We considered the possibility of $y\_i$ being contaminated with noise. In part of the analysis in Section 6, we assume this noise would be Gaussian, but except for that, there is no assumption on the type of output noise.

---

> > ### Comment · Reviewer_CYXN · 2023-08-18
> >
> > Thanks much to the authors for the detailed reply. Most of my questions are addressed. For the relation with Lasso, it seems the value of $\lambda^\prime$ will additionally depend on $\|y\|_1$, which might require more effort to tune compared with the standard Lasso framework. Also, other reviewers raise some concerns that I did not think of before. Overall, I tend to keep my previous rating.

---

> > > ### Author Response · Authors · 2023-08-20
> > >
> > > Again thank you very much for all the comments and the careful review.
> > >
> > > A small clarification: in the case of adversarial training you tune  $\delta$ which might be easier to adjust than the standard Lasso parameter since it does not depend on the noise amplitude (see Section 6). The value $\lambda'$ would not be accessed by the user of the method and would not be adjusted directly.

---

### Author Rebuttal · Authors · 2023-08-08

We sincerely thank all reviewers for taking the time to read and reflect on our paper. We are pleased to have received a generally positive response and that all the reviewers highlight the clarity of the paper and that it provides insight into adversarial training. We also believe the feedback provides us with useful information which will allow us to improve our paper.

---

### Decision · Program_Chairs · 2023-09-21

**Decision:**

Accept (spotlight)

**Comment:**

The paper provides a set of interesting results about the minimizers of the adversarial training objective when the underlying problem is linear regression (with using several regularization methods). As main messages, it is shown that adversarial training yields the minimum-norm interpolating solution in the overparameterized regime (more parameters than data), as long as the maximum disturbance radius is smaller than a threshold. And, conversely, the minimum-norm interpolator is the solution to adversarial training with a given radius. Also, it is shown that adversarial training can be equivalent to parameter shrinking methods (ridge regression and Lasso).

All the reviewers (and myself) are very positive about this paper. I believe that the results will be useful for the broad robust-ML community.

As a small note: I think the paper could also discuss the following papers as related works: (1) Dan et al on guarantees for adversarially robust gaussian classification, and (2) Dobriban et al on adversarially robust classification; Both papers have interesting results on classification which connect to linear classifiers (and also connect to some of the objectives considered in the paper).